# Efficacy of Double-Dose Dapsone Combination Therapy in the Treatment of Chronic Lyme Disease/Post-Treatment Lyme Disease Syndrome (PTLDS) and Associated Co-infections: A Report of Three Cases and Retrospective Chart Review

**DOI:** 10.3390/antibiotics9110725

**Published:** 2020-10-22

**Authors:** Richard I. Horowitz, Phyllis R. Freeman

**Affiliations:** 1HHS Babesia and Tick-borne Pathogens Subcommittee, Washington, DC 20201, USA; 2Hudson Valley Healing Arts Center, Hyde Park, NY 12538, USA; research@hvhac.com

**Keywords:** Lyme disease, post-treatment Lyme disease syndrome (PTLDS), dapsone combination therapy (DDS CT), double-dose dapsone combination therapy (DDD CT), babesiosis, bartonellosis, florescent in situ hybridization (FISH), persistent infection

## Abstract

Three patients with multi-year histories of relapsing and remitting Lyme disease and associated co-infections despite extended antibiotic therapy were each given double-dose dapsone combination therapy (DDD CT) for a total of 7–8 weeks. At the completion of therapy, all three patients’ major Lyme symptoms remained in remission for a period of 25–30 months. A retrospective chart review of 37 additional patients undergoing DDD CT therapy (40 patients in total) was also performed, which demonstrated tick-borne symptom improvements in 98% of patients, with 45% remaining in remission for 1 year or longer. In conclusion, double-dose dapsone therapy could represent a novel and effective anti-infective strategy in chronic Lyme disease/post-treatment Lyme disease syndrome (PTLDS), especially in those individuals who have failed regular dose dapsone combination therapy (DDS CT) or standard antibiotic protocols. A randomized, blinded, placebo-controlled trial is warranted to evaluate the efficacy of DDD CT in those individuals with chronic Lyme disease/PTLDS.

## 1. Introduction

Lyme disease affects over 300,000 Americans per year [1,2], and at least 2 million individuals in the United States have been estimated to be suffering from post-treatment Lyme disease syndrome (PTLDS) [3]. In Europe, Lyme borreliosis is also the most common tick-borne disease [4], and worldwide estimates suggest an increase in tick-vectored disease incidence and distribution [5]. Ticks can contain a broad range of bacteria (e.g., *Borrelia* spp., *Rickettsia* spp., *Francisella tularensis*), viruses (e.g., tick-borne encephalitis virus, Powassan virus), and parasites (babesia) [6]. Estimates from the World Health Organization suggest that 17% of human global infectious diseases are vector-borne, with *Borrelia burgdorferi* sensu lato complex and relapsing fever borreliosis comprising the major *Borrelia* spp. vectored by ticks [7]. Based on the geographical spread and increasing number of individuals suffering from Lyme and associated tick-borne diseases (TBDs), and significant health care costs associated with treatment failures [8,9], the necessity of finding effective treatments for Lyme borreliosis and associated co-infections is vitally important from a public health perspective.

Approximately 10–20% of individuals treated for Lyme disease with a 2–4-week course of antibiotics will go on to experience chronic, persistent fatigue, musculoskeletal pain, and neurocognitive difficulties that persist for more than 6 months, known as PTLDS [10]. The etiology of chronic Lyme disease/PTLDS is unknown, although several major hypotheses have been proposed to explain persistent symptoms, including persistence of *Borrelia* and/or borrelial antigens, persistent tick-borne co-infections, immune dysregulation, altered neural networks with central sensitization, and/or overlapping sources of inflammation [11,12,13,14]. The ability of *Borrelia* to persist in the body has been hypothesized to take place through multiple mechanisms. These include immune evasion with *Borrelia* changing its surface antigenic expression in response to host immune responses [15,16], persistence in the intracellular compartment [17,18], and change of morphological forms in various environments [19,20,21,22], resulting in atypical cystic forms [23], pleomorphic round bodies (cell wall deficient, L-forms) [20], as well as “persister” and “biofilm” forms [24,25,26,27,28]. The stationary, persister, and biofilm forms of *Borrelia burgdorferi* (Bb) have been found to be resistant to standard antibiotic treatments and a cause of persistent inflammation [29,30,31]. This phenotypic plasticity of *Borrelia* and its survival in biofilms may help to explain in part clinical conundrums and persistent symptomatology [32].

There have been several studies to date evaluating persister drugs and biofilm agents in the treatment of Lyme disease. Most of these have been in vitro studies, using essential oils, herbal compounds like Stevia, or drugs found through a search of the NCI compound collection or FDA approved drug library [29,33,34,35]. Two of these compounds, dapsone and disulfiram, which are both sulfa drugs, have been found to be effective against stationary phase *B. burgdorferi* [36,37,38] and evaluated in clinical studies. Disulfiram was found in a small case series to have a positive clinical effect in three patients who required intensive open-ended antimicrobial therapy for chronic relapsing neurological Lyme disease and relapsing babesiosis [39]. Dapsone combination therapy (DDS CT) has been found in two, separate retrospective case series, totaling 300 patients, to have a positive effect on eight major Lyme symptoms and improve treatment outcomes among patients with chronic Lyme disease/PTLDS and associated coinfections in those failing traditional antibiotic therapy [14,40].

Success in the prior DDS CT trials was operationally defined as improvement in percentage of normal after 6 months on DDS CT, and failure was operationally defined as remaining the same or worsening of the percentage of normal after at least 6 months of DDS CT. “Of 181 participants who gave both pre-DDS and DDS percentage scores, 14 participants reported feeling worse currently than they did before the DDS, 22 participants reported no difference, while all other participants (145) currently reported a higher percentage of normal” [14]. Causes of potential failures of DDS CT highlighted in Part 1 of our Precision Medicine study included evidence of chronic persistent infection with *Borrelia*, *Bartonella*, and *Mycoplasma* species, as well as *B. microti*. These were all shown to persist despite commonly prescribed courses of antibiotics or antimalarial/*Babesia* therapy [14]. Persistence of bacteria can be explained in part by bacterial biofilms, in which cells are protected from the immune system by surface exopolymers with polysaccharides [41]. Antibiotics have been shown in this model to kill regular cells, leaving dormant persisters alive, and when the concentration of antibiotic drops, they resuscitate and repopulate the biofilm [42].

Since the dosage of dapsone in the initial two studies varied between 25 and 100 mg/day, and the effect of ‘persister’ drugs like dapsone may depend on drug-dependent concentrations and their effects on biofilms [43], we decided to try a higher dose of dapsone twice a day (100 mg BID) for 1 month in several patients with a history of chronic, persistent relapses. We present here a total of 40 patients, including three case studies of individuals who took 7–8 weeks of dapsone combination therapy, using hydroxychloroquine, cimetidine, nystatin, a tetracycline, rifampin, and dapsone (DDS CT). Patients signed informed consent forms that listed the major side effects of dapsone combination therapy, which included Herxheimer reactions, anemia secondary to folic acid inhibition (hemolytic anemia was minimized by ensuring all patients had normal levels of G-6-P-D, or they were ineligible for the trial), rashes (secondary to sulfa sensitivity), and/or methemoglobinemia (secondary to increased oxidative stress and decreased heme oxygen carrying capacity). Patients were instructed to get regular laboratory testing with a complete blood count (CBC), comprehensive metabolic profile (CMP), and methemoglobin levels once at 100 mg of dapsone and to repeat laboratory testing weekly during the 2nd month of higher-dose therapy. Any major changes in their symptoms were to be reported immediately to the first author via an emergency cell phone number. After stopping all antibiotic therapy once the trial was completed, all 40 individuals remained on biofilm agents, folic acid replacement, and probiotics for the next several months. All three patients described in the case presentations remained in remission for time periods ranging between 2 and 3 years with no further Lyme and tick-borne symptoms after completing DDD CT. A retrospective chart review of an additional 37 patients demonstrated that, in total, 98% of patients improved their tick-borne symptoms post DDD CT despite some having active co-infections (*Babesia*, *Bartonella*), with 45% remaining in remission for 1 year or longer.

## 2. Case Presentations

### 2.1. Case 2.1

This 20-year-old African American male, with a past medical history significant for Lyme disease and babesiosis, as well as obsessive-compulsive disorder (OCD), became our patient in June 2017. His chief complaints included unexplained fevers, sweats, chills and/or flushing, significant fatigue, upset stomach, neck stiffness and cracking, headaches, hair loss (secondary to trichotillomania), blurry vision, and disturbed sleep with problems both falling asleep and early awakening. He had a tick bite at age 4 years old, where his ELISA and Western blot were positive, and he was treated by an infectious disease specialist for the next 6 years. During that timeframe, he was continuously on rotations of hydroxychloroquine (Plaquenil) and clarithromycin for 6 months, followed by rotations of hydroxychloroquine and a tetracycline for 6 months. Each time he tried coming off the antibiotics he had a relapse of his underlying symptoms and was therefore left on continuous antibiotic therapy during that 6-year time frame. At age 12, due to his lack of progress, his parents took him to see a pediatric Lyme disease specialist, who diagnosed him with both Lyme disease and babesiosis (*Babesia* titers were positive), since he still had ongoing fevers, sweats and chills, air hunger, fatigue, headaches, and insomnia. He was rotated to atovaquone, azithromycin, and doxycycline and remained on this protocol for the next several years, until he plateaued without any further improvement in his symptoms. He then saw a third Lyme specialist, who placed him on doxycycline, sulfamethoxazole/trimethoprim (Bactrim DS) with fluconazole, and atovaquone/proguanil (Malarone) for a short period of time, due to his ongoing fevers, sweats, and chills consistent with ongoing babesiosis. These treatments helped his symptoms, but as per prior regimens, were insufficient to maintain his health and allow him to remain in school.

During his first visit, he had been off antibiotics for several months, and all of his underlying tick-borne disease symptoms were relapsing. Before he saw us, his prior physician had put him on cefpodoxime proxetil, an oral third-generation cephalosporin, which was ineffective. He described moderate fevers, sweats, chills, severe fatigue (which increased postprandially), hair loss, upset stomach, joint pain, headaches, neck stiffness, blurry vision, and insomnia. He denied any other significant past medical history, social history, or family history, and his review of systems was otherwise unremarkable, except for occasional nocturia three times per night and frequent unexplained skin rashes. Physical examination revealed a well-developed, well-nourished African American male in no apparent distress. Sitting blood pressure was 129/80 with a pulse of 73 bpm, and standing 9 min, his blood pressure dropped to 119/83 with a pulse rate of 108 bpm. The 10-point drop in systolic blood pressure and 35-point increase in pulse rate was consistent with severe postural orthostatic tachycardia syndrome (POTS). The rest of his physical examination was unremarkable, except for some evidence of trichotillomania due to his underlying OCD.

We performed the following assessments: complete blood count (CBC), comprehensive metabolic profile (CMP), C-reactive protein (CRP), erythrocyte sedimentation rate (ESR), hormone levels (thyroid functions, DHEA/cortisol, pregnenolone), antigliadin antibody/tissue transglutaminase (TTG), IgE food allergy panel, immunoglobulin levels and subclasses, mineral levels (magnesium, copper, zinc), streptococcal titers (anti-streptomycin O (ASO), anti-DNase), viral titers (herpesvirus 6 (HHV-6), Epstein–Barr virus (EBV), cytomegalovirus (CMV)), *Stachybotrys* titer, urinalysis, and EKG. We also performed a comprehensive tick-borne panel, which included a C6 ELISA peptide; IgM/IgG Lyme Western blot; *Babesia microti* immunofluorescent assay (IFA); *Babesia duncani* antibody; *Babesia* fluorescent in situ hybridization (FISH) test; *Bartonella* titer and PCR; vascular endothelial growth factor (VEGF); Rocky Mountain spotted fever (RMSF, *Rickettsia rickettsii*) and tularemia (*Francisella tularensis*) titers; Q fever (*Coxiella burnetii*) titers; and *Mycoplasma*, *Chlamydia pneumoniae*, and *Brucella* titers. Significant results included evidence of multiple food allergies, consistent with leaky gut; low plasma copper (0.64 µg/mL, normal range between 0.80 and 1.75 µg/mL); prior exposure to HHV6 (antibody titers 1:1280, normal less than 1:80); and an elevated VEGF at 134 pg/mL (normal range 0–115 pg/mL) consistent with possible bartonellosis, with evidence of prior exposure to Lyme disease (positive 31, 39, and 41 kDa bands on an IgG Western blot). Urinalysis revealed mucus threads and calcium oxalate crystals, without any history of kidney stones.

We discussed the results of the physical examination and blood tests with the patient and his family. He was placed on a high-salt diet with 2 L of fluid per day for evidence of POTS, a strict hypoglycemic diet avoiding food allergens for his postprandial energy swings, and a different rotation of *Babesia* medication for his constant night sweats. He was given clindamycin 300 mg, 2 PO BID, Bactrim DS 1 BID, Malarone 2 PO BID, artemisinin 1 PO TID, grapefruit seed extract 2 PO BID, and nystatin 500,000-unit tablets 2 PO BID, along with Stevia extract 15 drops twice a day and triple probiotics, including *Saccharomyces boulardii* to prevent antibiotic-associated diarrhea.

Follow-up examination 2 months later revealed mild progress. He still complained of severe fatigue, irritable bladder, neck pain, stiffness of the neck and back, muscle pain, pain in the bottom of his feet which would come and go with shin pain (increased with sports), occasional lightheadedness, and mood swings with disturbed sleep, although his fevers and night sweats had improved on this regimen. On physical examination, his blood pressure dropped 20 points systolic (116/72 dropped to 96/70) with a 10-point increase in pulse rate at 10 min standing, consistent with ongoing POTS. Due to lack of adequate improvement, azithromycin 250 mg PO BID was added to his protocol, 4 days in a row per week, to extend the coverage against Bb, *Babesia*, and possible *Bartonella*, along with rifampin, 300 mg, 2 capsules PO BID one day per week. Mineral replacement was administered for the history of low copper and gastrointestinal support (glutamine, short-chain fatty acids, probiotics) was added for his history of leaky gut and food allergies.

Follow-up consultation on November 2017 revealed less joint pain and an improvement in *Babesia* symptoms (sweats and ‘air hunger’ were better) on clindamycin, azithromycin, and Bactrim along with Cryptolepis, but fatigue was increased on the weekend while doing pulsed rifampin, implying active intracellular infections and/or biofilm forms of *Borrelia*. Clindamycin was therefore stopped and changed to doxycycline, 150 mg PO BID, with rifampin, azithromycin, and Bactrim, and follow-ups between January and March 2018 showed slow improvement. Severe fatigue was primarily related to use of rifampin, and/or being off his hypoglycemic diet, but *Babesia* symptoms continued to improve with decreased sweats.

As of May 2018, all symptoms were slowly improving with multiple intracellular antibiotics, but he still complained of resistant fatigue, insomnia, neck pain, headaches, mild night sweats, and hair loss secondary to trichotillomania and ongoing OCD. The patient and his family did not want him to go on a selective serotonin reuptake inhibitor (SSRI) like paroxetine, so we discussed other options to treat his resistant tick-borne symptoms. Since the patient had been on almost continuous antibiotics since age 4 (14 years) and had never taken a persister drug regimen like dapsone, we discussed a trial of double-dose dapsone combination therapy (DDD CT) for 2 months based on prior published research for DDS CT. The patient and parents signed a consent form, which explained the protocol in detail. His glucose 6 phosphate dehydrogenase level (G6PD) was within normal limits, minimizing the risk of hemolytic anemia; sulfa-induced rashes would be unlikely, as he had been on Bactrim DS (sulfamethoxazole/trimethoprim) without side effects. The DDD CT protocol consisted of the following: 2 months of doxycycline 150–200 mg PO BID, rifampin 300 mg PO BID, hydroxychloroquine 200 mg PO BID, nystatin 500,000-unit tablets 2 PO BID, cimetidine 400 mg PO BID, and gradually increasing doses of dapsone: 25 mg week 1; 50 mg week 2; 100 mg weeks 3 and 4; and 100 mg of dapsone PO BID for 4 weeks, taken with leucovorin 25 mg PO BID month 1, 25 mg TID month 2, along with L-methyl folate 15–20 mg PO BID to help prevent significant anemia. Methylene blue, 50 mg PO BID, would be used during month 2 if there was any evidence of methemoglobinemia (a signed consent form had been given, explaining potential side effects and contraindications), and folic acid dosing would be increased for any significant increase in anemia. Nutritional support included three biofilm agents at the following doses: Stevia (NutraMedix) 15 drops PO BID; oregano oil capsules 60 mg PO BID; and Biocidin, two sprays in a glass of water. N-acetylcysteine (NAC) 600 mg PO BID, alpha lipoic acid 600 mg PO BID, and liposomal glutathione 1000 mg PO BID were used to reduce the risk of methemoglobinemia and severe Herxheimer reactions, with triple probiotics PO BID taken at least 1 h away from antibiotics. A typed flare protocol for severe, persistent Herxheimer reactions was also given, using PRN sodium bicarbonate, up to 2 g twice a day, or drinking fresh squeezed lemons/limes twice a day through a straw (to alkalize the body), along with 2000 mg of glutathione 2-3 times per day as needed. Laboratory values with a CBC, CMP, and methemoglobin levels were checked twice during month 1 and weekly during month 2 to rule out significant anemia and/or methemoglobinemia or detect any changes in liver or kidney function. Once the 2-month protocol was finished, the patient was to stop all antibiotics and only continue on folic acid supplementation, biofilm agents, and probiotics.

The patient and his family returned for an in-office consultation at the end of July 2018. He had finished the double-dose dapsone protocol approximately 1 month prior. He denied any significant Herxheimer reactions during treatment and had significant improvement in all of his baseline symptoms. Energy levels had significantly improved, neck pain and headaches had decreased, there were no noticeable night sweats, insomnia had improved, and cognition was within normal limits. He had some baseline mild right hip and foot pain, which was primarily increased playing soccer. He stated that post DDD CT, “it was the best he had felt in years” and his overall percentage of normal improved from 75% to 90% functioning. Baseline white cell counts (WBC) of 3.9–4.3 × 10^3^/uL remained stable during treatment and hemoglobin/hematocrit levels (H/H) of 14.2/43.5 dropped to 12.2/37.9 during month 2 (a 2 g drop in hemoglobin) with methemoglobin levels rising from 2.6% in month 1 to a maximum of 4.4% in month 2, which subsequently decreased to 3.9% and then 1.2% by the end of therapy (normal levels of methemoglobin range between 0% and 2.9%). He denied any symptoms of methemoglobinemia during treatment, using glutathione and methylene blue (no increased fatigue, headaches, shortness of breath, and/or blue hands/feet), and by the end of July 2018, his anemia had completely resolved, with an H/H of 15.9/48, increased from baseline values.

We repeated *Borrelia burgdorferi*, *Babesia*, and *Bartonella* testing post treatment. The Lyme ELISA was negative, his 31 kDa (Osp A) turned negative on the IgG Western blot, *Babesia microti* and *Babesia duncani* titers were negative, *Bartonella henselae* and *Bartonella quintana* titers were negative, and VEGF levels decreased to normal, but he had a positive *Babesia* FISH, consistent with active babesiosis, despite denying clinical symptoms of babesiosis. He therefore remained off all antibiotics and was only placed on Malarone (atovaquone/proguanil 100/250 mg) 2 PO BID with liposomal Artemisia 1 PO TID for several months with traditional Chinese herbs (Coptis, HH, circulation P). Six months post DDD CT, during a repeat consultation, he was at 100% of normal functioning. He denied fatigue, neck pain or headaches, joint pain, fevers, sweats, chills or air hunger, cognitive difficulties, or insomnia, and his OCD symptoms resolved with a decrease in trichotillomania. He was therefore tapered off Malarone and remained on mineral supplementation for his low copper, along with a hypoglycemic diet, avoiding food allergens. Repeat VEGF on December 2018 increased from less than 31 pg/mL to 250 pg/mL (normal range 31–86 pg/mL), but follow-up studies 6 months later showed that his VEGF returned to normal range with a negative *Babesia* FISH.

By December 2019, the patient was 1.5 years off of antibiotics and in remission post DDD CT, and he only complained of fatigue his 1st year of college if he was up late studying without adequate sleep. According to his mother, he was frequently sick during his childhood, and now he was never ill. Repeat adrenal testing showed phase 3 adrenal dysfunction with low morning and evening cortisol levels, which were contributing to his resistant fatigue. He was therefore placed on adaptogenic herbs and an adrenal glandular supplement which reversed any problems with energy/stamina. He was able to play recreational sports and walk a minimum of 2 miles per day. He denied any Lyme, *Babesia*, or *Bartonella* symptoms, his OCD and trichotillomania completely resolved, and his POTS/dysautonomia improved with mild tachycardia standing with no further drops in blood pressure. As of his last follow-up in August 2020, he had gone 2 years without a relapse of tick-borne symptoms (25 months) and was able to stay off antibiotics for the first time in 14 years.

### 2.2. Case 2.2

A 39-year-old Caucasian female presented to our medical office for the first time in July 1995 with a long and complex medical history significant for Lyme disease and Multisystemic Infectious Disease Syndrome (MSIDS), with more than 16 different overlapping etiologies discovered during a period of 2 decades which were contributing to her illness. These included Lyme disease (ELISA positive (1.20 IgG/IgM serology, normal < 1.0), CDC positive IgM Western blot, serum *Borrelia burgdorferi* PCR positive); babesiosis (positive *Babesia microti* titer 1:160); ehrlichiosis (*Ehrlichia chaffeensis*, human monocytic ehrlichiosis (HME) titers 1:160 positive); anaplasmosis (*Anaplasma phagocytophilum*, human granulocytic anaplasmosis (HGA) titers 1:160 positive); Rocky Mountain spotted fever (RMSF) IgG; enzyme immunoassay (EIA) positive (reference range negative/positive); exposure to *Bartonella henselae* (titers 1:64 positive); positive *Mycoplasma pneumoniae* titers with a serum *Mycoplasma fermentans* PCR positive (despite years of intracellular antibiotic rotations, including tetracyclines, rifampin, azithromycin, and/or quinolones); prior viral exposure to EBV and HHV-6; hormonal dysfunction with phase 2 adrenal dysfunction on a DHEA/cortisol test; hypothyroidism (low T3, free T3, elevated TSH); hypoglycemia/metabolic syndrome (HbA1c 5.9%, hyperlipidemia with an LDL 221); low vitamin D (18.0 ng/mL, normal range 31–100 ng/mL); and menopause with low estrogen, progesterone, and testosterone levels. She also had a history of gluten sensitivity (anti-gliadin antibody positive (39 units, negative values less than 20 U), tissue transglutaminase (TTG) negative); multiple food allergies (milk, eggs, nuts, wheat, soy, corn, shrimp, beef, and pork) with a history of candidiasis and leaky gut; inflammation (antinuclear antibody (ANA) positive, C4a (1416.7, normal range < 650)) with evidence of mast cell activation disorder (MCAD), proven by elevated levels of histamine (38 nmol/L, normal < 8) and chromogranin A (415 ng/mL, normal < 95) in the blood, which would result in migraines and gastrointestinal distress with dietary indiscretions; black mold exposure (positive *Stachybotrys* titers, black mold found growing on beams in the basement under her bedroom and bathroom, requiring remediation); and heavy metal exposure (elevated levels of mercury (7.7 mcg/day, normal < 5), cadmium (2.6 mcg/d, normal < 2), and aluminum (87 mcg/d, normal < 50) on a 6-h urine dimercaptosuccinic acid (DMSA) challenge) with elevated levels of mercury on hair analysis (7.8, normal < 1.2), along with detoxification problems compounded by low serum glutathione levels and deficiencies in serum mineral levels (red blood cell (RBC) magnesium, iodine, zinc). She also had a history of post-traumatic stress disorder (PTSD) secondary to a prolonged history of family trauma; postural orthostatic tachycardia syndrome (POTS)/dysautonomia (baseline BP 94/54 with a pulse rate of 80 beats per minute (BPM), which would decrease to 80/54 standing with a 20 point increase in pulse rate) with episodes of dizziness and pre-syncopal episodes; sleep apnea; and abnormal immunoglobulins (IgG 3 subclass deficiency (29 mg/dL, normal range 41–129) and elevated levels of IgM (303 mg/dL, normal range 40–230)). These various medical problems were addressed during a time period of over 20 years. Each time an abnormality on the 16-point MSIDS map was addressed (i.e., multiple infections, environmental toxins, detoxification issues with mineral deficiencies, hormonal dysfunction, POTS/dysautonomia, hypoglycemia, *Candida*, histamine intolerance with MCAD, sleep apnea, PTSD) clinical improvement was noted, but relapses were common, within days to weeks, especially when treatment for her tick-borne diseases was discontinued.

Her chief complaints consisted of severe fatigue; day sweats; night sweats and chills; neck stiffness and headaches with intermittent migraines; migratory joint and muscle pains in the neck, fingers, and toes; hair loss; rare tingling of the extremities; intermittent gas, nausea, vomiting, and constipation; blurry vision; balance problems with intermittent dizziness; moderate memory and concentration problems with poor organizational skills; poor sleep secondary to sleep apnea; and periods of sadness with depression and anxiety. Medication regimens used during a time frame of over 2 decades to address her Bb, *Babesia*, *Bartonella*, and *Mycoplasma* exposure according to chart review included rotations of doxycycline, atovaquone/proguanil and mefloquine; atovaquone and azithromycin; amoxicillin/clavulanic acid; cefuroxime axetil and azithromycin; amoxicillin and azithromycin; cefuroxime axetil, hydroxychloroquine and azithromycin; cefdinir, probenecid, telithromycin and hydroxychloroquine; and rotations of double intracellular antibiotic regimens with hydroxychloroquine (minocycline + azithromycin, minocycline + ciprofloxacin, minocycline + rifampin, doxycycline + moxifloxacin), each for several months at a time, along with a course of IV ceftriaxone 2 g/day for 1 month, with intermittent use of IV glutathione, which was effective for underlying symptoms and Herxheimer reactions. As of November 2017, post rotations of antibiotics for her tick-borne diseases (TBDs), she remained on Armour thyroid 60 mg/day; liothyronine 10 mcg BID; fludrocortisone 0.1 mg/day; midodrine 10 mg TID; Bioidentical hormone therapy (Bi-Est) with estradiol (0.25 mg), progesterone (40 mg), DHEA (5mg), testosterone (2.5 mg), pregnenolone (10 mg), cetirizine (10 mg), and famotidine (20 mg); occasional courses of fluconazole for *Candida* flare-ups; and low-dose naltrexone 4.5 mg HS.

During the above 20+ year medical history, when each of the above abnormalities were treated, she would often feel better with improved levels of functioning, but never maintain her health if anti-infective therapies against *Borrelia* and co-infections were stopped. Relapses of underlying tick-borne symptoms would often happen within days of stopping her various antibiotic protocols, requiring anti-infective herbal protocols to help stay in longer periods of remission (Zhang Traditional Chinese Medicine (TCM, Coptis/circulation P/HH) or the Cowden protocol with Samento/Banderol/Parsley/Burbur). There was also a significant increase in underlying symptoms of fatigue, headaches, myalgias, and cognitive difficulties if there were dietary indiscretions off of a strict hypoglycemic/*Candida*/mast cell diet. This would invariably lead to significant fatigue, gastrointestinal distress, and migraines. Her *Babesia* symptoms did however eventually resolve after several courses of atovaquone and a macrolide, or tetracycline, mefloquine, and atovaquone/proguanil, and her percentage of normal functioning kept increasing over the years, as she was rotated through multiple cell wall/cystic/double intracellular antibiotic regimens.

Due to frequent relapses within weeks off antibiotics, dapsone combination therapy (DDS CT) was instituted for the first time in 2015 after signing informed consent on potential side effects, using minocycline 100 mg PO BID, rifampin 300 mg PO BID, hydroxychloroquine 200 mg BID, dapsone 50 mg/day, nystatin 500,000 units BID, leucovorin 25 mg BID, L-methyl folate 15 mg/day, along with triple probiotics, and Stevia and Biocidin for biofilms. There was a significant improvement in baseline symptoms, with improvement in fatigue, pain, headaches, and cognitive difficulties. Only mild anemia (H/H 11/33.3) without any evidence of methemoglobinemia was noted. The regimen was subsequently stopped after the patient felt better, stabilizing after 6 months on the protocol; within several months off medications, symptoms relapsed again, with significant fatigue and brain fog. A *Borrelia burgdorferi* PCR was sent out January 2016 to IgeneX laboratories (Palo Alto, California) and returned positive, confirming ongoing active infection.

Since she reported symptom improvement on lower-dose dapsone (50 mg/day), the patient was placed back on DDS CT with minocycline, rifampin, and a higher dose of dapsone, increasing the dose from 75 mg/day to eventually 100 mg/day. The patient noticed an immediate improvement with the higher dapsone dose, with improvement in fatigue, pain, and cognitive functioning. As the H/H dropped to 9.7/28.6 after 1 month at 100 mg of dapsone while on leucovorin 25 mg BID and L-methyl folate 15 mg, with a mild increase in liver function tests (LFTs: ALT 39, normal range < 32 IU/L), folic acid dosing was increased to 25 mg of L-methyl folate, and the patient was given extra liver support (milk thistle, N-acetyl cysteine). The subsequent H/H improved to 10.7/31.4 and LFTs returned to normal. Blood methemoglobin levels rose to 3.7% in April 2016 while on 100 mg of dapsone (normal range < 1.5%), but the patient denied any symptoms of increased fatigue, headaches, shortness of breath, or cyanotic extremities, which would have been suggestive of symptomatic methemoglobinemia. Glutathione dosing was therefore increased to 500 mg PO BID for the elevated levels of methemoglobin, and several months later, while also on higher doses of folic acid, blood methemoglobin levels dropped to 2.5%, subsequently returning to normal range (0.6%) with her H/H improving to 11.1/32.9. The patient stayed on this DDS CT protocol for 6 months and stopped it at the end of 2016 when she was feeling close to 100% of normal functioning.

May 2017, having remained off antibiotics for 6 months, the patient again began to relapse with typical Lyme symptoms, although not as severely as with prior episodes. She complained of increased fatigue, headaches, neck pain/stiffness, paresthesias, and moderate to severe cognitive difficulties, despite strict compliance with diet and addressing all other abnormalities on the 16-point MSIDS map. A Lyme Western blot showed an increase in the 31 kDa band (Osp A). *Babesia* symptoms however had not returned post rotation of prior antimalarial medications (no day sweats, night sweats, flushing, unexplained cough, or air hunger), and *Babesia* titers returned to normal with negative *Babesia* FISH testing. *Bartonella henselae* titers also came down to normal over time, with negative PCRs and normal levels of vascular endothelial growth factors (VEGFs). It was therefore decided to try a double-dose dapsone protocol (DDD CT) for 2 months, after signing an informed consent. She gradually increased the doses of dapsone month 1 (up to 100 mg/day) and used dapsone 100 mg BID month 2, with the same antibiotic combinations as before (minocycline, rifampin, hydroxychloroquine, leucovorin, L-methyl folate, biofilm agents, and probiotics). Doses of glutathione were increased month 2 to 1000 mg BID to help reduce methemoglobin levels, with an increase in folic acid dosing to leucovorin 75 mg/day with 45 mg of L-methyl folate once baseline levels of H/H decreased.

The patient immediately felt better going back on DDS CT November 2017, with all symptoms improving at the end of month 1. H/H dropped from 13.1/38.9 at the beginning of therapy to 10.1/29.2 after 1 month, with methemoglobin levels remaining within normal limits (WNL) at 0.9%. Beginning the 2nd month of higher-dose dapsone therapy (DDD CT), she experienced a Herxheimer reaction for 3–4 days, with an increase in fatigue and cognitive difficulties, after which she felt 100% back to normal functioning, remaining at that level for the duration of treatment. Laboratory values towards the end of month 2 on dapsone 100 mg BID showed that methemoglobin levels remained within normal limits (WNL) at 1.2% but her H/H gradually dropped to 8.8/26.9. As there was no significant dyspnea except walking up steep hills, the regimen was continued until completion of the 8-week course, increasing folic acid doses as noted above during month 2. One month later, off dapsone, the H/H improved to 10.1/30.4; at month 2 post DDD CT, her H/H was back into normal range at 12.4/37.8. Folic acid dosing was subsequently discontinued, and she remained on probiotic and biofilm support for the next several months. As of August 2020, 30 months post DDD CT, the patient remained well without any relapse of Lyme and tick-borne symptoms, for the first time in over 25 years.

### 2.3. Case 2.3

A 50-year-old Caucasian female with a past medical history significant for Lyme disease, babesiosis, Graves’ disease with hyperthyroidism, hypertension, hyperlipidemia, obesity with insulin resistance and non-alcoholic fatty liver (NASH), chemical sensitivity, supraventricular tachycardia requiring cardiac ablation, left knee arthroscopic surgery for a torn meniscus, status post MVA with neck torsion, ADD, and bilateral cataract surgery presented to our office for a Lyme consultation in July 2010. She had a history of at least 10 tick bites and spent much of her life in heavily Lyme disease and *Babesia* endemic areas such as Martha’s Vineyard. Approximately 10 years ago, she began to develop multiple symptoms including significant fatigue, night sweats, joint pain, swollen glands, neck pain and stiffness, headaches, palpitations, shortness of breath, anxiety, depression, and insomnia. She went to see a local endocrinologist and was diagnosed with hyperthyroidism. She was placed on methimazole, which controlled her abnormal thyroid functions, but her health never returned to normal. One year later, her local physician diagnosed her with Lyme disease based on her symptoms and a CDC positive IgM and IgG Lyme Western blot. She was placed on doxycycline 100 mg PO BID and felt much better. She then went to see a local infectious disease specialist, who placed her on tetracycline HCL and hydroxychloroquine, but she did not feel well on this regimen and returned to her local physician to go back on doxycycline. She completed 4 months of antibiotic therapy and achieved between 70% and 80% of normal functioning. She then was rotated to azithromycin and rifampin for ongoing resistant symptoms but did not feel as well with this protocol, and she went to see a physician specializing in Lyme disease for a second opinion. Her testing returned positive for *Babesia* and *Borrelia burgdorferi*, and she was rotated to atovaquone and azithromycin. On this protocol her sweats got better, but otherwise there was a steady regression of her baseline underlying symptoms. She was therefore switched to amoxicillin 5 to 6 weeks before our initial consultation, titrating the dose upwards. Although she initially responded well, with a significant decrease in symptoms for the first 2 weeks, she then had a steroid injection in her neck for her chronic pain and severely relapsed; this was followed by a chemical exposure with hair straightening products, which also significantly worsened her symptomatology. At the time of our initial consultation, she felt much worse before starting her Lyme disease and *Babesia* therapy and was functioning at 10% of her normal functioning.

Chief complaints during the initial history and physical were fevers, sweats, chills, and flushing; weight gain; fatigue; sore throat and swollen glands; unexplained menstrual irregularity; loss of libido; upset stomach with constipation; chest pain/rib soreness with shortness of breath and an unexplained cough; palpitations; migratory joint pain in the knees, neck, hips, and ankles with neck and back stiffness; myalgias; headaches with intermittent migraines; tingling, numbness, burning, or stabbing sensations of the extremities; blurry vision; tinnitus and increased motion sickness; dizziness with poor balance; tremors; confusion with difficulty thinking; difficulty with concentration or reading; forgetfulness with poor short-term memory; disorientation and getting lost, going to the wrong places; difficulty with speech and writing; mood swings with irritability, anxiety, and depression; insomnia with early awakening; and exaggerated symptoms from alcohol use. Review of systems also revealed problems with a chronic postnasal drip, occasional wheezing, early satiety, and stress incontinence. Medication included methimazole 5 mg BID, amoxicillin 875 mg, 3 PO once daily (QD), Imitrex PRN, nystatin 500,000 units 3 PO QD, an estradiol patch, and progesterone 100 mg HS, as well as various vitamin and mineral supplements.

Physical examination was unremarkable except for an enlarged thyroid gland, consistent with Graves’ disease, and some left elbow tenderness on palpation. Laboratory testing done during the initial consultation included assessing thyroid functions; testing DHEA/cortisol; measuring vitamin and mineral levels; evaluating for heavy metal exposure (mercury, lead, arsenic, cadmium, aluminum); checking for food allergies; and expanding her tick-borne panel for exposure to Q fever, RMSF, and tularemia, as well as checking for exposure to *Bartonella*, *Chlamydia pneumoniae*, and *Mycoplasma pneumoniae*. Heavy metal testing returned showing elevated levels of lead (25 µg/g creatinine, normal range < 2) and mercury (17 µg/g creatinine, normal range < 4), and adrenal testing revealed phase 2 adrenal dysfunction with low cortisol in the morning (0.20 ng/mL, normal range 1–8.0 ng/mL) and at 2 p.m. (0.46 ng/mL, normal range 1–8.0 ng/mL). She was therefore placed on dietary restrictions, and told to decrease intake of larger fish, along with alpha lipoic acid 600 mg PO BID, and low-dose hydrocortisone in the morning and at noon (5 mg PO BID) with adaptogenic herbs (B vitamins, ashwagandha, rhodiola). These measures improved her energy/stamina. Since she had a significant improvement during the first 2 weeks on amoxicillin therapy before her steroid injection, she was placed on benzathine penicillin (Bicillin LA) 1.2 million units twice a week, hydroxychloroquine 200 mg PO BID, clarithromycin XL 500 mg PO BID, and atovaquone/proguanil 1 PO QD, along with probiotic support. Low-dose naltrexone (LDN) 2 mg HS was added for anti-inflammatory effects, and she was also started on bupropion (Wellbutrin XL) 150 mg PO BID for depression. She responded well to the Bicillin injections and oral antibiotics and remained on this therapy for the next several months. She also was eventually started on valsartan/hydrochlorothiazide for hypertension, rosuvastatin for hyperlipidemia (total cholesterol 317, triglycerides 253, HDL 47, LDL 219), and metoprolol XL 25 mg 1 QD for palpitations (a 24-h Holter monitor revealed frequent premature ventricular contractions (PVCs) without supraventricular tachycardia (SVT). A stress echocardiogram was done for her chest pain and cardiac risk factors, which showed no ischemic changes, and a Doppler/ultrasound of the carotids showed no evidence of plaque or stenosis. A low-carbohydrate, paleo-style diet was prescribed for significant weight gain (5 ft 7 in., 206 lbs) with hyperinsulinemia (70.7 uIU, normal < 24.9 uIU), hyperuricemia (uric acid 8.2. normal range < 6.0 mg/dL), elevated hs-CRP (3.7 mg/L, normal < 1), and a CT abdomen suggestive of a fatty liver.

Over the following several years, due to ongoing resistant symptoms, the patient was rotated through multiple antibiotic regimens including various oral cephalosporins (cefuroxime axetil, cefdinir), metronidazole, doxycycline, minocycline, azithromycin, rifampin, and quinolones (ciprofloxacin, levofloxacin, moxifloxacin). Although there were temporary improvements in symptomatology with these protocols, as well as improvements using rotations of antimalarial therapy for babesiosis for ongoing sweats (lumefantrine/artemether (Coartem), clindamycin), she would relapse with an increase in her baseline symptoms each time she was taken off therapy. In May 2017, her primary care physician decided to institute a trial of IV ceftriaxone, 2 g QD, along with DDS CT, for increased *Borrelia* specific banding on her Western blot (23 kDa, Osp C) accompanied by resistant symptoms of fatigue, joint pain, and severe ongoing memory/concentration problems off treatment.

She was placed back on DDS CT with minocycline 100 mg PO BID, pulse rifampin 300 mg, 2 PO BID one day/week (for GI tolerance), and dapsone, slowly increasing doses to 100 mg/day with leucovorin 25 mg BID and L-methyl folate 15 mg/day. N-acetyl-cysteine (NAC), alpha lipoic acid (ALA), and liposomal glutathione (GSH) were given twice a day for inflammation and detoxification support with triple biofilm agents (Serrapeptase, monolaurin, Stevia) and triple probiotics. Rifabutin 150 mg PO BID eventually was used instead of rifampin, with better GI tolerance. She improved with this protocol, and by June 2017, 1 month later, she had significantly less fatigue and musculoskeletal pain and improved cognition, functioning at 90% of normal. Cognitive problems were still present, but slowly improving, and IV glutathione (GSH) was highly effective for both Herxheimer reactions and resistant brain fog, relieving her symptomatology within 15 min of an IV infusion. Her PCP therefore left her on the same protocol until she decided to go to Germany for hyperthermia treatment in early August 2017.

She was taken off her DDS CT by the German physicians and left on IV ceftriaxone and metronidazole, with low-dose niacin, while being given IV ozone and stem cells during her visit. She tolerated the treatment well and felt an improvement after the second course of hyperthermia. Upon returning to the United States, despite jet lag, her joint pain and flexibility as well as her cognition had improved, so we removed the PICC line, stopped all antibiotics, and administered 2 months of mitochondrial support with glycosylated phospholipids (NT Factors), CoQ10, and acetyl-L-carnitine, adding nicotinamide adenine dinucleotide (NADH) if fatigue were to persist. During an in-office follow-up 2 months later, she was still gradually improving, only complaining of mild fatigue, mild cognitive problems, and neck pain. *Babesia* symptoms were gone, and a repeat *Babesia* titer was negative. Repeat adrenal function with a DHEA/cortisol test also showed that she no longer suffered from adrenal fatigue, even off low-dose hydrocortisone.

In December 2017, after 4–5 months off IV and oral antibiotics and post hyperthermia treatment, she began to have a relapse of underlying symptoms. These included fatigue, flushing, increased neck pain, blurry vision, and ongoing memory/concentration problems. There were also new, unexplained allergy symptoms that were arising, with an intermittent red face and swollen lips. She went for an allergy evaluation, and we checked an IgE food panel and antibodies against the alpha gal allergen, which were all negative. Magnetic resonance imaging (MRI) of the neck did not reveal any structural cause for her neck pain, and an eye examination was performed for her blurry vision, which was WNL. Since the patient’s symptoms were not yet severe, she decided to try an alternative treatment regimen with her PCP, using alpha and beta thymosin injections, which gave some relief of symptoms during a 3-month trial. The flushing also resolved off niacin, and she was given fenofibrate (Tricor) 145 mg instead for her hypertriglyceridemia, but this was stopped 3 days later secondary to a possible allergic reaction with ongoing facial erythema.

The patient had a telemedicine consult April 2018, since she continued to slowly relapse 9 months off antibiotics. She had gone from 90% down to 60% of normal functioning, with increasing fatigue, neck pain with neuralgia on the right side of the neck, head pressure, and increased brain fog. Since her G6PD levels were WNL (239, normal range between 46–376 U) with a normal CBC (H/H 14.5/42.2) and no baseline elevations in methemoglobin (0.3% normal range < 1.9%), we discussed a trial of DDD CT, starting at the beginning of May 2018. After signing an informed consent, she began doxycycline 200 mg PO BID, rifampin 300 mg PO BID, gradually increasing doses of dapsone until at 100 mg BID, hydroxychloroquine 200 mg PO BID, nystatin 500,000 units 2 PO BID, cimetidine 400 mg PO BID, and leucovorin 25 mg BID with 15 mg of L-methyl folate during month 1, increasing L-methyl folate to 25 mg PO BID month 2. This was along with triple biofilm agents and triple probiotics with NAC, ALA, and GSH. A CBC, CMP, and methemoglobin levels were drawn every 2 weeks during the 1st month of treatment and weekly during the 2nd month of DDD CT, at which point she would stop the therapy and remain on biofilm agents, folic acid, and probiotics.

By July 2018, the patient had completed DDD CT, having missed the last week of treatment secondary to gastrointestinal upset. She was functioning at 90% of normal and reported feeling the best she had felt in the past 20 years. Her CBC had slightly decreased on dapsone while remaining on high-dose folic acid (H/H 13.6/40.5), and there was no significant methemoglobinemia on glutathione 1000 mg twice a day (methemoglobin 0.3%, normal < 1.9%). Fatigue and nerve pain had completely resolved, generalized arthralgias and myalgias were gone, except for some residual neck pain, moods were significantly better with decreased anxiety, and brain fog was mild, but improved. As of November 2018, she remained at a higher level of functioning than before doing DDD CT, although she had slipped from 90% to 85% of normal functioning. Her anxiety was returning along with palpitations, mild fatigue, neck stiffness, bladder dysfunction with urgency and frequency (history of a dropped bladder), and difficulty concentrating and reading. Laboratory values 6 months post DDD CT in December 2018 were also WNL (H/H 14.4/42.7) with a normal creatinine (0.9, normal < 0.95) and normal liver functions, despite the history of fatty liver (AST 26, normal < 34 U/L; ALT 37, normal < 36).

As of March 2019, most of her baseline Lyme symptoms had resolved, but she still complained of some fatigue and daily morning headaches, which would promptly resolve with 400 mg of ibuprofen. She was placed on a stricter hypoglycemic diet for her hyperinsulinemia and occasional low blood sugars on her CMP (glucose 62). By August 2019, the patient felt she was in complete remission, at 100% of her normal functioning, with no further Lyme symptoms. Only residual neck stiffness remained, which was felt to be due to her status post MVA with multiple neck traumas falling off horses. Her fatigue and headaches had resolved with a stricter hypoglycemic diet, and her urinary and bowel problems (constipation) also improved post total abdominal hysterectomy, for fibroids that were compressing her bladder and colon. It was the best she had ever felt since falling ill 20 years ago. As of August 2020, she had remained off antibiotics for over 2 years, with no further relapse of symptoms of chronic Lyme disease/PTLDS.

These three patients represent a snapshot of the effects of DDD CT therapy. In order to demonstrate the potential for this treatment to improve long-term tick-borne symptoms, we completed a retrospective chart review of an additional 37 patients for a total of 40 patients. The regimen was offered to approximately 100 patients, and we chose 40 patients who had completed the protocol at least 1 year prior in order to evaluate their remission and improvement status. All of the patients in this chart review had been ill for at least 1 year and had been treated by multiple health-care providers, failing traditional antibiotic therapy for Lyme disease, which included but was not limited to tetracyclines, macrolides, penicillins, and cephalosporins. Patients were given a standardized protocol sheet after determining eligibility. The DDD CT care plan is shown in Figure 1.

## 3. Methods

We examined 40 charts of patients who completed the DDD CT protocol and were off antibiotics for 1 year or longer. We assessed co-infection status, age, length of illness, and response to treatment, i.e., improvement, remission, or lack of response to the DDD CT protocol. Remission was defined as resolution of all active tick-borne symptoms for 1 year or longer. All 40 patients in our retrospective chart review met the criteria for a clinical diagnosis of Lyme disease supported by a physician-documented erythema migrans (EM) rash and/or positive laboratory testing, including a positive ELISA/enzyme immunoassay (and/or C6 ELISA), immunofluorescent antibody (IFA), Centers for Disease Control and Prevention (CDC) positive IgM and/or IgG Western blot (WB), PCR, *Borrelia*-specific bands (23, 31, 34, 39, 83/93) on a WB [14], and/or positive ELISpot (lymphocyte transformation test (LTT)). These patients had either failed or had an inadequate response to prior antibiotic therapy and/or had relapsed with persistent symptoms after stopping anti-infective therapy.

Institutional Review Board approval was not required for this research since this was a retrospective review of patients’ charts who were previously undergoing treatment for tick-borne illness under the care of the first author. After signing informed consent forms that outlined the proposed benefits and potential risks of therapy, patients volunteered to enroll in a preliminary DDD CT trial at our medical center based on the drug’s action on persister and biofilm forms of the bacteria. Patients under the age of 18 years, having a known allergy to DDS or any medication used in the trial, and/or having significant laboratory abnormalities including a pre-trial anemia or G6PD deficiency were excluded from our study.

## 4. Results

Of 40 participants, 21 were male and 19 were female. Age range was between 20 and 84 years old (M = 46.45, SD = 15.85). Fifty-five percent were less than 50 years old, and the remaining 45% were older than 50. Sixteen patients (40%) had been ill for longer than 20 years, 10 patients (25%) had been ill for 10–20 years, 9 patients (22.5%) had been ill between 5 to 9 years, and 5 patients (12.5%) had been ill between 1-4 years. Thirty percent of patients (N = 12) had a history of EM rashes and thereby met the criteria for PTLDS. Thirty-seven out of 40 patients (92.5%) had at least one co-infection, 12 patients (30%) had two co-infections, and 5 patients (12.5%) had three or more co-infections. Twenty-one patients (53%) were *Babesia microti* antibody positive, 5 patients (13%) were *Babesia duncani* antibody positive, 6 patients (15%) were *Babesia* FISH positive, 5 patients (13%) were *Ehrlichia* antibody positive, and 3 patients (8%) were *Anaplasma* antibody positive. Eighteen patients (45%) were *Bartonella* antibody positive (*B. henselae, B. quintana*), 1 patient had a *Bartonella* skin biopsy positive by immunofluorescence, 1 patient was *Bartonella* PCR positive, 7 patients (17.5%) were *Bartonella* FISH positive, and 6 patients (15%) had evidence of elevated VEGF, an indirect marker of active *Bartonella*. Co-infection status and treatment results are presented in Table 1.

Treatment results: Thirty-nine of 40 patients (98%) showed improvement of their tick-borne symptoms, and no patients had a worsening of symptoms post therapy. Forty-five percent had a resolution of all active Lyme symptoms post treatment for 1 year or longer if there was no evidence of active co-infections. Seven out of 12 patients (58%) with an EM rash and history of PTLDS remained in remission, and the remainder, 42%, showed a moderate improvement in underlying symptoms (37%) above their baseline functioning. Among six patients who were *Babesia* FISH positive, three (50%) remained in remission, and the other three patients improved (50%). Among 11 patients with a history of *Bartonella* exposure, with no evidence of active infection (biopsy negative, PCR negative, FISH negative), 5 (45%) remained in remission, and 6 (55%) improved their underlying symptomatology. Among five out six patients with evidence of active *Bartonella* (elevated VEGF) who were *Bartonella* FISH negative, two (40%) remained in remission and three improved, while none of the patients who were *Bartonella* FISH positive (seven patients) or PCR/biopsy positive (two patients) went into remission. Further examination of those patients revealed that among those who were *Bartonella* FISH positive, six out of seven (86%) improved, and one had no change in symptoms, while patients who were PCR or biopsy positive for *Bartonella* had an improvement in symptoms that ranged from 10% to greater than 30% above baseline functioning.

Age and length of illness was also evaluated to determine effect on treatment outcome. Among 22 out of 40 patients less than 50 years old, 9 patients went into remission, 12 patients improved, and 1 patient had no change in symptoms. In those 18 patients greater than 50 years old, 9 patients were in remission and 9 improved. Regarding the length of illness, 16/40 patients were ill for 20+ years and 5 went into remission, while 11 improved; 10 patients were sick between 10 and 20 years, and 5 remained in remission, while 5 improved; 9 patients were ill between 5 and 9 years, and 7 went into remission, while 2 improved; and finally, 5 patients were sick for 1-4 years, where 1 patient went in remission, 3 improved, and 1 had no change in symptoms. The only patient who did not improve on the DDD CT protocol had exposure to both species of *Babesia* (*B. microti*, *B. duncani*), *Anaplasma*, and both species of *Bartonella* (*B. henselae*, *B. quintana*), with a positive *Bartonella* FISH test post therapy.

## 5. Discussion

Chart review of 40 patients who were at least 1 year out after completing DDD CT revealed the importance of several factors. Age did not appear to be a major variable in determining treatment outcome in this small retrospective study. Regarding length of illness, most patients had been ill for a long period of time, and a length of illness less than 10 years in duration appeared to lead to better remission outcomes. This is consistent with data collected from other published studies that tracked patient progress over time. Regarding the presence of EM rashes and treatment outcome, 100% improved, and 58% with a history of PTLDS remained in remission. The remainder of patients (42%) showed a moderate improvement in underlying symptoms (37%) above their illness functioning. This positive response to DDD CT in PTLDS patients implies that active infections may be playing an important role in driving chronic symptomatology.

There has been a longstanding medical debate regarding the etiology of chronic Lyme disease/PTLDS, with two published standards of care for the diagnosis and treatment [44,45]. Current IDSA treatment guidelines (2006) have relied on four randomized, controlled trials (RCTs) for persistent disease and advised against retreatment [44], although earlier National Institutes of Health (NIH) placebo-controlled, double-blinded RCTs of PTLDS showed some benefit on primary or secondary outcome measures in two out of the four trials using IV ceftriaxone [46]. A further biostatistical review of antibiotic retreatment in all four Lyme disease RCTs was therefore performed to understand the discrepancies [47]. DeLong et al. concluded that the design assumptions in the two Klempner trials were unrealistic and underpowered to detect meaningful treatment effects; that the Krupp trial was well designed, finding statistically significant and meaningful improvement in fatigue; and that the Fallon trial corroborated this finding, while also demonstrating improvement in cognitive functioning at week 12, which declined by week 24 [47]. Although improvement in fatigue and cognitive functioning was noted in two of the four trials, sustained improvement was lacking, requiring new treatment strategies.

Other authors have similarly noted relapses in Lyme symptoms post discontinuation of antibiotics and long-term impairment of functional status [48]. Although case reports and uncontrolled trials have reported the efficacy of prolonged antibiotic therapy [49,50,51], symptoms have often been shown to relapse after discontinuation of therapy [52]. Relapse in underlying Lyme symptoms with persistent fatigue, musculoskeletal pain, neuropathy, and neurocognitive/neuropsychiatric difficulties can potentially be explained by multiple etiologies. These include persistence of spirochetal antigens/peptidoglycans in the joints post treatment leading to Lyme arthritis [12]; persistent infection with *Borrelia burgdorferi* [14,53,54,55], due to invasion of the spirochete into the joints [56,57], ligaments [58], eyes [59,60], central nervous system [61,62], and protected niches including the intracellular compartment [17,18] and fibroblasts [63,64]; persistent tick-borne co-infections [14], including *Babesia* [65,66,67,68,69] and associated co-infections, i.e., *Bartonella* [70,71], *Mycoplasma* spp. [72,73]; and viral infections such as human herpes virus 6 (HHV-6) [14,74], as well as other medical problems causing overlapping sources of inflammation with downstream effects [13]. These include up to 16 factors identified on the MSIDS model [75,76], including immune dysfunction/immune deficiency [13,15,77], environmental toxins with detoxification problems [78], GI problems, food allergies [79,80], nutritional deficiencies [81,82], hormone and autonomic nervous system dysregulation [83,84,85], mitochondrial dysfunction [86,87], neuropsychiatric problems, and/or sleep disorders [13,88,89,90]. All patients reported here had evidence of multiple overlapping etiologies accounting in part for resistant symptoms, including evidence of associated co-infections (*Babesia*, *Bartonella*, *Mycoplasma fermentans*), hormonal dysfunction, food allergies and leaky gut, environmental toxin exposure with mineral deficiencies, POTS/dysautonomia, neuropsychiatric problems, and sleep disorders. Patient 2 also had evidence of persistent *Borrelia* and *Mycoplasma* species by serum polymerase chain reaction (PCR) despite years of targeted antibiotic therapy. Although addressing these multiple etiologies led to clinical improvements, it was not however until DDD CT therapy was instituted, using a higher dose persister drug regimen with dapsone 100 mg PO BID along with a tetracycline, rifampin, and biofilm agents, that all three of our case studies were finally able to stay in remission for prolonged periods of time (24–30 months). Among the other 37 patients who took DDD CT, 98% improved and no patient had a worsening of symptoms, with 45% of all patients enrolled in the study remaining in remission.

In the past several years, university researchers have determined that *B. burgdorferi* may change morphological forms in different environments [20,21], forming cystic forms (L-forms, S-forms, round bodies, cell-wall-deficient forms) [91,92], and metamorphosing into drug-tolerant persister and biofilm forms [24,25,27,28]. Persisters are known to occur in a broad range of bacterial infections, leading to dormancy and persistent infection [22,30,93,94]. These persister cells escape the effects of antibiotics without genetic modification, do not grow in the presence of antibiotics, and become a significant fraction of cells in the stationary phase and in biofilms [95]. Persister forms and biofilm formation potentially play an important role in antibiotic resistance and reoccurrence of Lyme disease (*B. burgdorferi*) and neuroborreliosis with other *Borrelia* species including *Borrelia afzelii* and *Borrelia garinii* [28,29,96]. Moreover, the importance of biofilms has been identified in other chronic, resistant infections. These include biofilm formation with *Pseudomonas aeruginosa* in patients with cystic fibrosis, *Haemophilus influenza* and *Streptococcus pneumoniae* in chronic otitis media, and enteropathogenic *Escherichia coli* and *Klebsiella* in recurrent urinary tract infections [97,98]. Biofilm formation has also been shown to play a role in *Candida* infections in chronic periodontitis [99] and in parasitic infections with *Pneumocystis* spp. in lung tissue [100], as well as posing a risk to medical implants, such as central venous catheters, heart valves, ventricular assist devices, coronary stents, neurosurgical ventricular shunts, and breast implants [101].

Innovative approaches for treating persister and biofilm forms of *Borrelia* have emerged during the past decade. New compounds with high activity against stationary-form *Borrelia burgdorferi* persisters have been found by searching the NCI compound collection [35], screening an FDA approved drug library [29], and identifying new drug candidates using high-throughput screening [37]. Potential candidates with efficacy against Bb persisters in culture included daptomycin, cefoperazone and doxycycline [102], sulfa drugs [36] including dapsone [40] and disulfiram [37,38], bee venom/mellitin [103], essential oils (oregano oil, cinnamon bark, clove) [33], and Stevia extract [34]. For in vitro biofilm forms of Bb [27], effective killing of Bb has been noted with novel herbal compounds such as Biocidin [104], Stevia [34], baicalein and monolaurin [105,106], and oregano oil [33]. However, only results for dapsone and disulfiram have been published in retrospective clinical trials to date [14,39,40]. In one study, three patients who had required open-ended antimicrobial therapy for symptoms of chronic, relapsing Lyme disease and babesiosis had a positive response to a finite course of disulfiram and remained clinically well without a relapse for a period of 6–23 months [39]. One patient relapsed on disulfiram and required retreatment. Dapsone combination therapy (DDS CT) has been found to be effective in TBDs in both a case study of a patient with Lyme disease, co-infections, and autoimmune disease [107] and in two retrospective clinical trials involving a total of 300 patients [14,40]. Eight major Lyme symptoms showed statistical improvement (*p* < 0.05) with DDS CT, including sweats, fatigue, musculoskeletal pain, neuropathy, headaches, cognitive difficulties, and sleep disorders. However, the dose and length of time on DDS CT varied in the two trials (from 25 mg to 100 mg/day), and relapses off DDS CT were not uncommon. The primary difference in this study of 40 patients with chronic Lyme disease/PTLDS was the 100 mg BID dose of dapsone the 2nd month, as compared to doses ranging up to 100 mg QD in our previous studies. The higher dose of dapsone was effective in keeping 45% of our patients in symptomatic remission for the first time in decades, with only 7–8 weeks of DDD CT leading to remission times between 25 and 30 months in our three case studies. This was significantly different than our experience of using lower doses of dapsone, which was effective but led to relapses upon discontinuation of therapy.

Prior studies done by Dr Eva Sapi and researchers at the University of New Haven showed that DDS CT (tetracycline, rifampin, dapsone) was highly effective against the biofilm forms of Bb and that higher concentrations of dapsone were more effective in lowering biofilm mass [43]. The major findings were that dapsone, as a single drug and in combination with doxycycline and doxycycline + rifampin, had the most significant effect in reducing the mass and viability of *B. burgdorferi* biofilm. In a repeat in vitro study of DDS CT by Horowitz and researchers at the University of New Haven [108], in order to further evaluate the effectiveness of the antibiotics on the attached biofilm form of *B. burgdorferi*, crystal violet biofilm, Baclight LIVE/DEAD viability, and dimethylmethylene blue (DMMB) glycosaminoglycan assays (GAG) were used to evaluate the amount of the protective layers of biofilm polysaccharide matrix before and after antibiotic treatments. The results showed that at 50 µM doses of dapsone (vs. 10 µM), the higher dapsone dose had the most significant effects on residual GAG amounts of *Borrelia* biofilm compared to the untreated control (*p* value < 0.01). These culture results support the clinical findings of the superiority of higher-dose dapsone therapy for chronic tick-borne symptoms.

*B. burgdorferi* biofilm, the most antibiotic-resistant form of the bacteria, has been shown to be present in borrelial lymphocytomas [28] and was recently found to be a dominant form in a human autopsy study from a Lyme disease patient [109]. The clinical significance of biofilm-like microcolonies was further highlighted in a recent animal study by Johns Hopkins researchers where the biofilm-like microcolony and stationary-phase planktonic forms (free cells) caused more severe Lyme arthritis and inflammation than actively growing log-phase spirochetes [31]. Dapsone and disulfiram have both been found to cause significant Herxheimer reactions in patients with chronic Lyme disease/PTLDS [14,39,40], consistent with biofilm/persister forms causing significant inflammation. The production of inflammatory chemokines and cytokines by *B. burgdorferi* is known to be a primary factor driving clinical symptomatology [110,111].

Based on the above in vitro and our in vivo studies, there is a need for safe and effective drugs that can eliminate all morphological forms of *B. burgdorferi* including persisters and attached biofilm forms. Both disulfiram and dapsone are known to have potential adverse effects. Disulfiram commonly causes fatigue, sleepiness, headaches, and a metallic taste, although more severe reactions including dermatological, hepatic (hepatitis, hepatotoxicity), cardiac, neurological (peripheral and/or axonal polyneuropathy, sensory-motor polyneuropathy, optic neuritis, seizures), and psychiatric (confusion, psychosis) reactions, may result [112]. In patients with Lyme disease and associated co-infections, underlying cardiac, hepatic, neurological, and psychiatric conditions can also co-exist [88,113,114,115,116,117], confounding etiologies and requiring a differential diagnosis with monitoring of symptoms and/or use of lower doses of disulfiram to minimize side effects [39]. Dapsone also has four common side effects, described as “Do No H.A.R.M.”, i.e., Herxheimer reactions (due to increased inflammatory cytokine production), anemia (secondary to inhibition of folic acid metabolism, or hemolysis due to G-6-P-D deficiency), rashes (due to sulfa sensitivity), and methemoglobinemia (due to increased oxidative stress and diminished oxygen-carrying capacity) [14,118,119,120]. Although some of these symptoms were seen in our patients undergoing treatment with DDD CT (Herxheimer reactions, anemia, mild elevations in methemoglobin), adverse side effects were minimized by ruling out G-6-P-D deficiency; using high-dose folic acid therapy with folinic acid (50–75 mg/day) and L-methyl folate (30–45 mg/day); and administering glutathione precursors (NAC 600 mg BID), alpha lipoic acid (ALA, 600 mg BID), and glutathione (GSH, 1000 mg BID) with methylene blue 50 mg BID as needed. Any decrease in red cell counts or significant anemia secondary to dapsone resolved in all of our case studies within 1–2 months of stopping DDD CT while remaining on folic acid supplementation, and none of our case studies developed rashes or significantly elevated levels of methemoglobin. Use of NAC, ALA, and GSH helped to decrease oxidative stress, support detoxification, and minimize the risk of methemoglobinemia [121,122,123], while doses of glutathione were increased to 2000 mg QD or BID along with alkalization (using sodium bicarbonate or fresh squeezed citrus) for Herxheimer reactions and/or any increased levels of methemoglobin [76,124,125]. Methylene blue can also be given orally to mitigate and rapidly reduce methemoglobinemia [120]. It was only necessary in one of our three case studies reported here, although in other chronically ill Lyme-MSIDS patients given dapsone at 100 mg or higher [14], oral methylene blue was occasionally needed and was effective in keeping methemoglobin levels below 5%, allowing continuation of therapy. Finally, high-dose probiotics (greater than 80 billion CFUs/day) with multiple strains of *Acidophilus*, *Bifidobacterium*, and *Saccharomyces boulardii* were effective in preventing antibiotic-associated diarrhea [126].

A 7–8-week regimen of DDD CT using the above persister and biofilm protocol with higher doses of dapsone was therefore found to be safe and effective in our 40 patients. It was superior to lower-dose dapsone combination therapy (DDS CT), leading to long-term remission in 45% of patients. However, several important questions remain regarding the safety and efficacy of persister drug regimens. Bacterial cells in biofilms have been shown to have increased antibiotic resistance compared to planktonic forms, leading to recalcitrant infections [127,128], and persister cells have been shown to retain their phenotype for days or weeks after withdrawal from colony–biofilm culture [129]. Although 7–8 weeks of treatment with DDD CT was effective in these 39/40 patients reported here, an important question that needs to be addressed is how different *Borrelia* species and/or associated co-infections including *Babesia* and *Bartonella* species would affect treatment outcomes in other patients with long-term tick-borne symptoms. Upon discontinuation of therapy, six out of seven of our patients who were *Bartonella* FISH positive showed improvement, but none reached remission of their symptoms. Fifty percent of patients who were *Babesia* FISH positive reached remission, and the other 50% reported improvement. Other studies have found active infection with *Babesia* and/or *Bartonella* spp. by FISH testing in chronically ill tick-borne patients [130,131]. Previously, dapsone and disulfiram were both found to be effective in decreasing symptoms of babesiosis, although relapses were noted with both medications [14,39]. Resistance to standard treatments has been reported for both *Babesia* and *Bartonella* spp. [69,70], and resistant biofilm and persister forms have also recently been reported for *Bartonella* [132,133]. Would addressing *Babesia* and/or *Bartonella* with newer medication regimens, including tafenoquine for *Babesia* [134] and/or novel combination therapies for *Bartonella* (i.e., macrodantin, rifampin, methylene blue, gentamycin with essential oils) [133], prior to DDD CT improve clinical outcomes in co-infected patients? Similarly, the three biofilm agents (Stevia, oregano oil, Biocidin) we used in our study were all found to have efficacy against biofilms and morphological forms of *Borrelia* [33,34,104], but would other biofilm agents or combinations against *Borrelia* and/or associated co-infections be more efficacious [33,135]? Randomized controlled trials (RCTs) would be necessary to answer these important questions.

## 6. Conclusions

The use of newer persister medications or biofilm agents was not evaluated in any of the four prior RCTs for the treatment of persistent symptoms of Lyme disease [136,137,138] or the PLEASE trial, which used doxycycline, clarithromycin, hydroxychloroquine, and IV ceftriaxone [139]. Similarly, the multiple overlapping etiologies on the 16-point MSIDS model causing inflammation with downstream effects, as highlighted in our three cases (including co-infections, environmental toxin exposure, mineral deficiencies, food allergies, mast cell activation, hormonal dysfunction, POTS/dysautonomia, and sleep disorders) [13], were not evaluated and addressed in prior RCTs. These factors may have interfered with the long-term success of antibiotic therapy in the Klempner, Krupp, and Fallon RCTs, where resistant symptoms and relapses were seen after discontinuation of therapy. Based on recently published research implicating the importance of biofilm and persister forms of Bb in chronic infection, the clinical superiority of a 7–8 week protocol of DDD CT leading to long-term remission in 45% of our patients for 1 year or longer without active co-infections, and the significant number of new patients contracting Lyme disease and PTLDS each year in the United States and worldwide [3], it is vital that an evaluation of these new persister protocols be performed in placebo-controlled, blinded, randomized clinical trials.

## Figures and Tables

**Figure 1 antibiotics-09-00725-f001:**
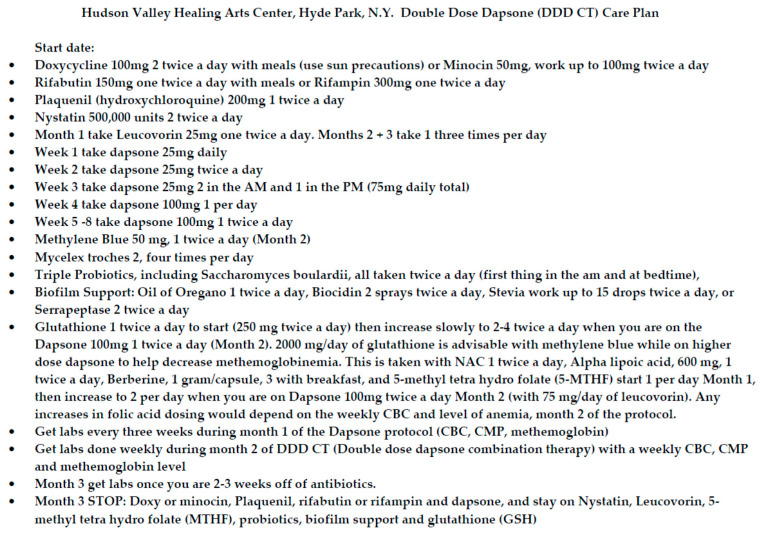
Double-dose dapsone combination therapy (DDD CT) patient care plan.

**Table 1 antibiotics-09-00725-t001:** Co-infection status and treatment response in 40 patients on DDD CT.

Response to Treatment	Bm 21	Bd 5	Bab FISH + 6	E 5	A 3	Bart AB + 18	VEGF ↑ 6	Bart PCR +/Biopsy + 2	Bart FISH + 7	1 Co-inf 20	2 Co-inf’s 12	3 Co-inf’s 5
Remission	12	2	3	2	1	6	2	0	0	8	6	2
Improved 10–20%	3	1	1	2	1	3	2	1	2	3	2	2
Improved 20–30%	2	0	1	0	0	2	2	0	3	5	1	0
Improved >30%	3	1	1	1	0	6	0	1	1	4	3	0
No change	1	1	0	0	1	1	0	0	1	0	0	1

Abbreviations: *Babesia microti* (Bm); *Babesia duncani* (Bd); *Babesia* florescent in situ hybridization (Bab FISH); Ehrlichia (E); Anaplasma (A); *Bartonella* antibody (Bart AB); vascular endothelial growth factor (VEGF); *Bartonella* polymerase chain reaction (Bart PCR); *Bartonella* florescent in situ hybridization (Bart FISH); co-infections (Co-inf).

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
