# Peer review of "Efficacy of Double-Dose Dapsone Combination Therapy in the Treatment of Chronic Lyme Disease/Post-Treatment Lyme Disease Syndrome (PTLDS) and Associated Co-infections: A Report of Three Cases and Retrospective Chart Review"

_antibiotics, 2020, doi:10.3390/antibiotics9110725_

Round 1
Reviewer 1 Report
This paper describes 3 patients with chronic symptomatology attributed to “chronic Lyme disease” and associated tick-borne infections and the change in their symptoms following administration of Dapsone.
The paper contains many misstatements. The very first sentence states “at least 2 million individuals in the United States have been reported to be suffering from Post-Treatment Lyme Disease Syndrome (PTLDS) [3]”. In fact, that reference describes a statistical simulation and provides the 2 million number as an estimated outer confidence limit of the prevalence, given the most extreme assumptions, and not “at least”. However, since the purpose of the paper is to describe 3 specific patients who received this medication, my comments will focus on these case reports.
- None of the 3 ever had typical Lyme disease symptoms. Hence the inclusion of ‘post treatment Lyme disease syndrome’ in the title and elsewhere is inappropriate
- There is no factual basis for the diagnosis of Lyme disease or other tick-borne infections in any of the 3.
- Case 1. We are told the Lyme ELISA and Western blot were positive at age 4 in this 20 year old. There is no description of possible exposure and these test values are not provided. Without these the validity of the original diagnosis cannot be assessed. None of the subsequent symptoms support the diagnoses. The only additional testing that is described is (line 158) 3 bands on an IgG Western blot. Since extensive testing for tick borne illnesses was obtained, and no additional results are provided, we can assume all other results were negative. Hence there is no evidence this patient ever had the listed diagnoses.
- Case 2: This patient had decades of symptoms, none of which were typical of Lyme or other tick-borne diagnoses, had no described exposure, and had elevated totak serum IgM. Diagnosis was based on Lyme “ELISA + [1.20 IgG/IgM serology, normal < 1.0], CDC positive IgM Western Blot, serum Borrelia burgdorferi PCR +), Babesiosis (positive Babesia microti titer 1:160), Ehrlichiosis (Ehrlichia chaffeensis, Human Monocytic Ehrlichia [HME] titers 1:160 +)”. In a patient with longstanding symptoms a low positive Lyme ELISA supported only by a positive IgM blot has an extremely high negative predictive value for the diagnosis of Lyme. Similarly, the other positive serologies are likely an artefact of the elevated total IgM. The significance of the positive serum PCR cannot be assessed without knowing which lab performed the test and that lab’s experience with false positives, as this finding is extraordinarily rare outside acute disease.
- Case 3: This patient, who had potential exposure, is said to have had Lyme disease in the past, but it is not clear if this refers to a diagnosis prior to her current decade of symptoms; if it predates the 10 year symptomatology, no validating data or clinical description are provided. Her current symptoms began a decade before the current treatment, and IgG and IgM Western blots were said to be positive with no further detail. Since there is no description of a positive ELISA, one can assume it was negative, making the Western blot meaningless. We are told that on subsequent evaluation (line 421) “Her testing returned positive for Babesia and Borrelia burgdorferi” but no detail is provided, again making the statement uninterpretable. She is described as having extensive subsequent testing but no results are provided; hence an assumption that it was all negative would be reasonable. None of her symptoms were diagnostic of Lyme disease and no data in support of other tick-borne illnesses is provided.
Author Response
Response to Reviewer 1:
- Line 29-30. Reviewer one, stated “The very first sentence states “at least 2 million individuals in the United States have been reported to be suffering from Post-Treatment Lyme Disease Syndrome (PTLDS) [3]”. In fact, that reference describes a statistical simulation and provides the 2 million number as an estimated outer confidence limit of the prevalence, given the most extreme assumptions, and not “at least”.
Response: We changed the word ‘reported’ to ‘estimated’ on line 30.
- “None of the 3 ever had typical Lyme disease symptoms.”
Response: We do not agree. The three cases studies reported here, describe symptoms of chronic LD/PTLDS reported by other authors in the published scientific literature. This includes:
- Nancy A. Shadick, MD, MPOH, Charlotte B. Phillips, MPH, Eric L. Logigian, MD, Allen C. Steere, MD, et al. The Long-Term Clinical Outcomes of Lyme Disease: A Population-Based Retrospective Cohort Study. Ann of Int Med. Oct 15, 1994. https://doi.org/10.7326/0003-4819-121-8-199410150-00002
“Results:
Compared with the control group (n = 43), the Lyme group (n = 38; mean duration from disease onset to study evaluation, 6.2 years) had more arthralgias (61% compared with 16%; P < 0.0001); distal paresthesias (16% compared with 2%; P = 0.03); concentration difficulties (16% compared with 2%; P = 0.03); and fatigue (26% compared with 9%; P = 0.04), and they had poorer global health status scores (P = 0.04). The Lyme group also had more abnormal joints (P = 0.02) and more verbal memory deficits (P = 0.01) than did the control group. Overall, 13 patients (34%; 95% CI, 19% to 49%) had long-term sequelae from Lyme disease (arthritis or recurrent arthralgias [n = 6], neurocognitive impairment (n = 4), and neuropathy or myelopathy [n = 3]). Compared with controls, patients who had long-term sequelae had higher IgG antibody titers to the spirochete (P = 0.03) and received treatment later (34.5 months compared with 2.7 months; P < 0.0001).
- Musculoskeletal and neurologic outcomes in patients with previously treated Lyme disease.
Shadick NA, Phillips CB, Sangha O, Logigian EL, Kaplan RF, Wright EA, Fossel AH, Fossel K, Berardi V , Lew RA , Liang MH . Annals of Internal Medicine [01 Dec 1999, 131(12):919-926].
“BACKGROUND: Previous follow-up studies of patients with Lyme disease suggest that disseminated infection may be associated with long-term neurologic and musculoskeletal morbidity. OBJECTIVE: To determine clinical and functional outcomes in persons who were treated for Lyme disease in the late 1980s. DESIGN: Population-based, retrospective cohort study. SETTING: Nantucket Island, Massachusetts. PARTICIPANTS: 186 persons who had a history of Lyme disease (case-patients) and 167 persons who did not (controls). MEASUREMENTS: Standardized medical history, physical examination, functional status measure (Medical Outcomes Study 36-item Short Form Health Survey [SF-36]), mood state assessment (Profile of Mood States), neurocognitive tests, and serologic examination. RESULTS: The prevalence of Lyme disease among adults on Nantucket Island was estimated to be 14.3% (95% CI, 9.3% to 19.1%). In multivariate analyses, persons with previous Lyme disease (mean time from infection to study evaluation, 6.0 years) had more joint pain (odds ratio for having joint pain in any joint, 2.1 [CI, 1.2 to 3.5]; P = 0.007), more symptoms of memory impairment (odds ratio for having any memory problem, 1.9 [CI, 1.1 to 3.5]; P = 0.003), and poorer functional status due to pain (odds ratio for 1 point on the SF-36 scale, 1.02 [CI, 1.01 to 1.03]; P < 0.001) than persons without previous Lyme disease”.
- Horowitz, R.I.; Freeman, P.R. Precision Medicine: retrospective chart review and data analysis of 200 patients on dapsone combination therapy for chronic Lyme disease/post-treatment Lyme disease syndrome: part 1. International Journal of General Medicine 2019:12 101–119.
“Surveys were conducted in office, online, and via telephone to gather patient information. The symptom questionnaire that the patients have completed is derived from a larger validated Lyme questionnaire (13) and the work of Shadick et al.(14) Symptom severity of eight symptoms was gathered both before beginning DDS CT and after 6 months of treatment, following a protocol from our previous study by Horowitz and Freeman on DDS.(8)
Symptoms measured were:
Fatigue and/or tiredness
Muscle and/or joint pain
Headache
Tingling and/or numbness and/or burning of extremities
Sleep problems
Forgetfulness and/or brain fog
Difficulty with speech and/or writing
Day sweats and/or night sweats and/or flushing
Participants rated the severity of these symptoms both before DDS CT and after 6 months, to examine whether DDS CT decreased the severity of these symptoms. Data were gathered using a 5-point severity scale wherein patients indicated the severity of each symptom, where 1 represented no symptom, and 5 represented the most severe.
Inclusion criteria
All 200 patients in our retrospective chart review met the criteria for a clinical diagnosis of Lyme disease supported by a physician documented erythema migrans (EM) rash and/or positive laboratory testing, including a positive ELISA/enzyme immunoassay, and/or C6 ELISA, immunofluorescent antibody (IFA), Centers for Disease Control and Prevention (CDC) positive IgM and/or IgG Western blot (WB), PCR, Borrelia-specific bands (23, 31, 34, 39, 83/93) on a WB,15 and/or positive ELISpot (lymphocyte transformation test [LTT]). These patients had either failed or had an inadequate response to prior antibiotic therapy and/or had relapsed with persistent symptoms after stopping anti-infective therapy.
Exclusion criteria
Patients under the age of 18 years, having a known allergy to DDS or any medication used in the trial, and/or having significant laboratory abnormalities including a pre-trial anemia were excluded from our study.
Surveys included
Surveys with questions regarding DDS status (currently or previously taking the drug), DDS dosage, coinfections, other drugs, side effects, and symptom severity for common Lyme symptoms before and after at least 6 months of DDS treatment (the full survey is available on request from the authors) were included.
Results
In order to analyze this, paired-samples t-tests were performed on each symptom with pre-DDS and DDS conditions of: fatigue and/or tiredness: t(164)=10.69, P<0.001, muscle and/or joint pain: t(164)=8.13, P<0.001, headache: t(164)=5.35, P<0.001, tingling and/or numbness and/or burning of extremities: t(164)=6.71, P<0.001, sleep problems: t(164)=6.17, P<0.001, forgetfulness and/or brain fog: t(164)=9.84, P<0.001, difficulty with speech and/or writing: t(164)=8.70, P<0.001, and day sweats and/or night sweats and/or flushing: t(164)=8.36, P<0.001.
A Wilcoxon signed-rank nonparametric test (which was run due to the small range of severity ratings) showed a statistically significant change in severity ratings of the same eight symptoms for pre-DDS and DDS conditions. Results indicated that for fatigue and/or tiredness: Z=−8.624, P<0.001, muscle and/or joint pain: Z=−7.295, P<0.001, headache: Z=−5.587, P<0.001, tingling and/or numbness and/or burning of extremities: Z=–7.302, P<0.001, sleep problems: Z=−6.363, P<0.001, forgetfulness and/or brain fog: Z=–8.169, P<0.001, difficulty with speech and/or writing: Z=−7.873, P<0.001, day sweats and/or night sweats and/or flushing: Z=–8.081, P<0.001.
These results further confirm that patients had a significant change in all eight chronic Lyme symptoms (Table 1).
- Empirical Validation of the Horowitz Multiple Systemic Infectious Disease Syndrome Questionnaire for Suspected Lyme Dis-ease. Maryalice Citera, Ph.D., Phyllis R. Freeman, Ph.D., Richard I. Horowitz, M.D., International Journal of General Medicine 2017:10 249–273. http://www.ncbi.nlm.nih.gov/pubmed/28919803
” Factor analysis: SPSS was used to conduct an exploratory factor analysis of the HMQ Symptom Checklist to examine its factor structure and construct validity. The HMQ Symptom Checklist items were factor analyzed using maximum likelihood estimation. An oblique rotation was used to identify factors. Because symptoms often co-occur, this allowed correlations among the factors.
We considered an item to load on a factor if it had a ≥0.25 loading. In cases where an item loaded on more than one factor at the level of ≥0.25, we considered it to load on the factor with its highest loading. In all cases, the primary loading made the most sense in terms of the factors and latent variables underlying the factors.
Six factors were identified by examining the scree plot, the Eigenvalues, and variance explained. As factor analysis involves both statistical analysis and judgment, the factor labels are based on interpretations. These were labeled neuropathy, cognitive dysfunction, musculoskeletal pain, fatigue, dysautonomia, and cardio/respiratory. The items and their factor loadings are listed in Table 2”.
Based on the above 4 referenced papers, the patients in our case studies did in fact have symptoms that have been associated with individuals treated for Lyme disease.
- “Hence the inclusion of ‘post treatment Lyme disease syndrome’ in the title and elsewhere is inappropriate”.
Response: Since we redid the paper, and added a retrospective review of 40 cases in total, 12 of whom had PTLDS (s/p EM rashes), we are now able to include PTLDS in the title and in the rest of the paper.
- “There is no factual basis for the diagnosis of Lyme disease or other tick-borne infections in any of the 3 cases”.
Response: Apart from the 12 EM rashes that are now described in our paper, all 40 met the following criteria:
See line 582-590 of the text, which was added to the paper, to clarify how the diagnosis of LD was made. The following statement was accepted and published in 3 or our prior peer-reviewed papers (Horowitz, R.I.; Freeman, P.R. Precision Medicine: retrospective chart review and data analysis of 200 patients on dapsone combination therapy for chronic Lyme disease/post-treatment Lyme disease syndrome: part 1. International Journal of General Medicine 2019:12 101–119
https://www.ncbi.nlm.nih.gov/pubmed/30863136
Horowitz, R.I.; Freeman, P.R. Precision Medicine: The Role of the MSIDS Model in Defining, Diagnosing, and Treating Chronic Lyme Disease/Post Treatment Lyme Disease Syndrome and Other Chronic Illness: Part 2. Healthcare 2018, 6, 129.
https://www.ncbi.nlm.nih.gov/pubmed/30400667
Horowitz RI, Freeman PR (2016) The Use of Dapsone as a Novel “Persister” Drug in the Treatment of Chronic Lyme Dis-ease/Post Treatment Lyme Disease Syndrome. J Clin Exp Dermatol Res 7: 345. doi:10.4172/2155-9554.1000345
“All 40 patients in our retrospective chart review met the criteria for a clinical diagnosis of Lyme disease supported by a physician documented erythema migrans (EM) rash and/or positive laboratory testing, including a positive ELISA/ enzyme immunoassay, and/or C6 ELISA, immunofluorescent antibody (IFA), Centers for Disease Control and Prevention (CDC) positive IgM and/or IgG Western blot (WB), PCR, Borrelia-specific bands (23, 31, 34, 39, 83/93) on a WB,15 and/or positive ELISpot (lymphocyte transformation test [LTT]). These patients had either failed or had an inadequate response to prior antibiotic therapy and/or had relapsed with persistent symptoms after stopping anti-infective therapy”.
- “Case 1. We are told the Lyme ELISA and Western blot were positive at age 4 in this 20 year old. There is no description of possible exposure and these test values are not provided. Without these the validity of the original diagnosis cannot be assessed. None of the subsequent symptoms support the diagnoses. The only additional testing that is described is (line 158) 3 bands on an IgG Western blot. Since extensive testing for tick borne illnesses was obtained, and no additional results are provided, we can assume all other results were negative. Hence there is no evidence this patient ever had the listed diagnoses.
Response: I contacted the parents of the 20-year-old, who confirmed the positive testing. Because the records are 16 years old, they could not access the older results, but provided a 2011 positive Western Blot. The family lived in New Jersey, in a highly Lyme endemic area, and three out of 4 family members have tested positive for Lyme and associated co-infections. As per lines 158-163, “Significant results included evidence of multiple food allergies, consistent with leaky gut; low plasma copper (0.64 ug/mL, normal range between 0.80-1.75 ug/ml); prior exposure to HHV6 (antibody titers 1:1280, normal less than 1:80), and an elevated VEGF at 134 picograms/milliliter (normal range 0-115 pg/ml) consistent with possible Bartonellosis, with evidence of prior exposure to Lyme disease (positive 31 kDa, 39 kDa and 41 kDa bands on an IgG Western blot)”. He had a positive ELISA, and the bands that are marked positive, i.e. Osp A (31) and 39 are highly significant bands on a Western blot, showing prior exposure. This has been reported previously in the medical literature, and it is important to remember, that Lyme disease is first and foremost a clinical diagnosis. This has been established by both the FDA and the CDC. The CDC Surveillance Case Definition is
- a) a case with EM or;
- b) a case with at least one objective manifestation such as meningitis, cranial neuropathy, arthritis, or AV block, that is laboratory confirmed. In the words of the CDC:
“This surveillance case definition was developed for national reporting of Lyme disease; it is not in-tended to be used in clinical diagnosis.” Centers for Disease Control Prevention MMWR56(23);573-576, June 15, 2007
http://www.cdc.gov/ncphi/disss/nndss/casedef/lyme_disease_2008.htm
How do we make that clinical diagnosis? The patient must have a reasonable history of tick exposure; have signs and symptoms consistent with the illness, and laboratory testing which helps confirm the diagnosis. Since MRI’s, SPECT scans and PET scans of the brain are not able to definitively deter-mine if a patient has neurological Lyme disease, physicians will occasionally perform a spinal tap and look at markers in the spinal fluid to determine if Borrelia burgdorferi has invaded the CNS.
Unfortunately, spinal taps also have their limitations. Although increased antibody production in the spinal fluid can be seen in early Lyme disease with a lymphocytic meningitis or encephalitis, in late stage neurological Lyme patients, patients can have normal cerebrospinal fluid (CSF) antibody studies. For example, Dr. Coyle and Dr. Schutzer studied 35 patients with specific Lyme Antigen (Osp A) in their cerebrospinal fluid, indicative of neurological Lyme disease. Of these patients studied, although the Lyme antigen was positive, 43% had no evidence of antibodies to Lyme in their CSF testing, and 47% had otherwise normal routine CSF analyses. 60% of these patients were also sero-negative for Lyme disease when tested with standard blood tests, implying that a patient can have Lyme disease despite a negative blood test and a negative spinal tap. The authors concluded that, “neurologic infection by B. burgdorferi should not be excluded solely on the basis of normal routine CSF or negative CSF antibody analyses.”
- Coyle PK, Schutzer SE, Deng Z, Krupp LB, Belman AL, Benach JL, Luft BJ. Detection of Borrelia burgdorferi-specific antigen in antibody-negative cerebrospinal fluid in neurologic Lyme dis-ease. Neurology. 1995 Nov;45(11):2010-5;
Patients may also be seronegative for Lyme disease because of sequestration of antibody in immune complexes.
- Schutzer SE, Coyle PK, Belman AL, Golightly MG, Drulle J. Sequestration of antibody to Borrelia burgdorferi in immune complexes in seronegative Lyme disease. Lancet. 1990 Feb 10;335(8685):312-5;
Patients will therefore not necessarily meet the CDC two step criteria to diagnose Lyme disease (a positive Elisa followed by a positive Western blot). Again, this surveillance case definition was developed for national reporting of Lyme disease, and was not intended to be used in clinical diagnosis:
http://wwwn.cdc.gov/NNDSS/script/casedef.aspx?CondYrID=752&DatePub=1/1/2011%2012:00:00%20AM
Why do patients fail two-tiered testing? The blood tests to diagnose Lyme are known to lack sufficient sensitivity and specificity to pick up all patients with the disease.
False seronegativity has been extensively reported in the peer review medical literature:
- Steere AC. Seronegative Lyme disease. JAMA. 1993 Sep 15;270(11):1369
- Kaiser R. False-negative serology in patients with neuroborreliosis and the value of employing of different borrelial strains in serological assays. J Med Microbiol. 2000
- Pikelj F, Strle F, Mozina M. Seronegative Lyme disease and transitory atrioventricular block. Ann Intern Med 1989 Jul 1;111(1):90. Oct;49(10):911-5.
- Dejmkova H, Hulinska D, Tegzova D, Pavelka K, Gatterova J, Vavrik P. Seronegative Lyme arthritis caused by Borrelia garinii. Clin Rheumatol. 2002 Aug;21(4):330-4.
- Brunner M. New method for detection of Borrelia burgdorferi antigen complexed to antibody in seronegative Lyme disease. J Immunol Methods. 2001 Mar 1;249(1-2):185-90.
- Breier F, Khanakah G, Stanek G, Kunz G, Aberer E, Schmidt B, Tappeiner G. Isolation and polymerase chain reaction typing of Borrelia afzelii from a skin lesion in a seronegative patient with generalized ulcerating bullous lichen sclerosus et atrophicus. Br J Dermatol. 2001 Feb;144(2):387-92.
- Schutzer SE, Coyle PK, Belman AL, Golightly MG, Drulle J. Sequestration of antibody to Borrelia burgdorferi in immune complexes in seronegative Lyme disease. Lancet. 1990 Feb 10;335(8685):312-5
- Dattwyler RJ, Volkman DJ, Luft BJ, Halperin JJ, Thomas J, Golightly MG. Seronegative Lyme disease. Dissociation of specific T- and B-lymphocyte responses to Borrelia burgdorferi. N Engl J Med. 1988 Dec 1;319(22):1441-6
Finally, even the FDA has stated“…a patient with active Lyme disease may have a negative test result…”Brown SL, Hansen SL, Langone JJ. (FDA Medical Bulletin) Role of serology in the diagnosis of Lyme disease. JAMA. 1999 Jul 7;282(1):62-6.
One way to help confirm the clinical diagnosis, after ruling out other diseases, is to look at the bands on the Western Blot. The significance of these bands on the Western blot was described in the peer review medical article by Ma et al: Serodiagnosis of Lyme Borreliosis by Western Immunoblot: Re-activity of Various Significant Antibodies against Borrelia burgdorferi. Journal of Clinical Microbiology, Feb. 1992, p. 370-376.
“The significance of various antibodies against Borrelia burgdorferi was studied by Western blot (immunoblot) by using 578 human serum samples. The proteins regularly detected by using samples from patients with Lyme borreliosis were those with bands with molecular masses of 94, 83, 75, 66, 60, 55, 46, 41, 39, 34, 31, 29, 22, and 17 kDa. The detectable frequencies of most of these proteins appeared to be significantly different between the sera from patients with Lyme borreliosis and those from normal control individuals as well as from the group with syphilis”
There are therefore 2 divergent standards of care regarding the testing and treatment of Lyme disease in the United States: The IDSA and ILADS guidelines. These 2 standards of care vary in their diagnostic and treatment recommendations. Key points of the IDSA guidelines are that Lyme tests are held to be reliable, and patients with persistent Lyme symptoms after standard treatment have Post Treatment Lyme Disease (“PTLD”) with possible autoimmune phenomenon driving chronic illness. The ILADS guidelines on the other hand state that Lyme tests are unreliable and that multiple factors may account for persistent symptoms. I was one of the authors of the evidence-based 2004 ILADS guidelines, published in Expert Review of Anti Infective Therapy (Evidence-based guidelines for the management of Lyme Disease. Cameron, Horowitz, et al. Expert Review of Anti-Infective Therapy 2(1) 2004).
Physicians in the United States have a choice to follow either of these two evidence-based guidelines, but there are known problems with the IDSA guidelines. There was a published scientific review in the Archives of Internal Medicine which analyzed the overall level of evidence behind the IDSA guidelines: Analysis of Overall Level of Evidence Behind Infectious Diseases Society of America Practice Guidelines, Dong Heun Lee, MD; Ole Vielemeyer, MD; Arch Intern Med. 2011;171(1):18-22
As per the conclusions of the authors: “We analyzed the strength of recommendation and overall quality of evidence behind 41 IDSA guidelines released between January 1994 and May 2010.
Their conclusions: “More than half of the current recommendations of the IDSA are based on level III evidence only (opinion). Until more data from well-designed controlled clinical trials become available, physicians should remain cautious when using current guidelines as the sole source guiding patient care decisions”.
The majority of physicians in the United States in fact do not follow IDSA guidelines. They treat for seronegative disease and treat for extended periods of time. An article that was published in the journal Infection in 1996 by Dr Sam Donta highlighted the discrepancy and showed that the majority of physicians do not treat according to IDSA guidelines: “For chronic Lyme disease, 57% of responders treat 3 months or more.” (Ziska MH, Donta ST, Demarest FC. Physician preferences in the diagnosis and treatment of Lyme disease in the United States. Infection 1996 Mar-Apr;24(2):182-6). A recent study by the CDC confirmed the same data. The CDC surveyed a representative sample of people in the US and found that only 39% of those with Lyme disease were treated in accordance with blanket short-term recommendations in the IDSA guidelines. The majority were treated for longer periods:
- Hook S, Nelson C, Mead P. Self-reported Lyme disease diagnosis, treatment, and recovery: Results from 2009, 2011, & 2012 Health Styles nationwide surveys. Presented at The 13th
International Conference on Lyme Borreliosis and other Tick Borne Diseases, Boston, MA Aug 19, 2013. Available from: http://archive.poughkeepsiejournal.com/assets/pdf/BK211780914.pdf.
The reason that the majority of physicians in the United States do not follow IDSA guidelines is be-cause more than half of their recommendations are based on level III evidence only, and we know that the two-tier testing approach recommended by the IDSA does not always work in clinical practice. According to these guidelines, an immunoblot is not to be performed if the ELISA is negative, despite the poor sensitivity of ELISA tests ranging from 34 to 70.5%.
The effect of using the IDSA guidelines would be to miss roughly half of those suffering with Lyme disease:
- Marangoni, A. et al. Comparative evaluation of three different ELISA methods for the diagnosis of early culture-confirmed Lyme disease in Italy. J. Med. Microbiol. 54, 361-367 (2005);
- Ang, C.W.,et al. T. Large differences between test strategies for the detection of anti-Borrelia antibodies are revealed by comparing eight ELISAs and five immunoblots. Eur. J. Clin. Microbiol. Infect. Dis. 30, 1027-1032 (2011).
- Wojciechowska-Koszko, et al. Serodiagnosis of borreliosis; Arch. Immunol. Ther. Exp. 59, 69-77 (2011).
John Hopkins University also found problems with the CDC two-tiered testing approach. In 2005, John’s Hopkins did a study and found that the CDC two-tiered testing missed up to 55% of positive Lyme cases (Coulter,et al.,J Clin Microbiol 2005;43:5080-5084). A NYS DOH Study done in 1996 which was reported to the CDC, found the number of patients missed by the two-tiered protocol (without an EM rash) to be even higher: 81% of Non-EM Cases were not confirmed with present two tiered testing algorithms (CDC correspondence with NYS DOH, April 15th, 1996).
Inaccurate diagnostic tests–based on technology that is over 20 years old-create medical uncertainty in both the diagnosis and treatment of Lyme disease
- Stricker RB, Johnson L. Lyme disease diagnosis and treatment: lessons from the AIDS epi-demic. Minerva Med. 2010 Dec;101(6):419-25
- Direct Diagnostic Tests for Lyme Disease. Steven E. Schutzer, et al. Oct 3, 2018. Clinical Infectious Diseases® 2018;XX(XX):1–6
The IgM Western blot has also found to be a frequent marker of exposure for Lyme disease, even with a negative ELISA and IgG Western Blot:
- Horowitz, R.I.; Freeman, P.R. Precision Medicine: retrospective chart review and data analysis of 200 patients on dapsone combination therapy for chronic Lyme disease/post-treatment Lyme dis-ease syndrome: part 1. International Journal of General Medicine 2019:12 101–119
https://www.dovepress.com/articles.php?article_id=44148
- Characteristics of seroconversion and implications for diagnosis of post-treatment Lyme dis-ease syndrome: acute and convalescent serology among a prospective cohort of early Lyme disease patients. Alison W. Rebman et al. Clinical Rheumatology, 30 May 2014
Understanding the role of laboratory testing in Lyme disease and other tick-borne diseases requires understanding that the 2-tiered protocol of using a Lyme ELISA followed by a Western Blot will miss approximately 1/2 of the patients secondary to the insensitivity of the ELISA test:
- Cook MJ, Puri BK. Commercial test kits for detection of Lyme borreliosis: a meta-analysis of test accuracy. Int J Gen Med. 2016 Nov 18;9:427-440. eCollection 2016.
- Cook MJ, Puri BK. Application of Bayesian decision-making to laboratory testing for Lyme disease and comparison with testing for HIV. International Journal of General Medicine 2017:10 113–123
The utility of the Western Blot is therefore based on understanding specific bands which reflect exposure to Borrelia: 23, 31, 34, 39, 83-93.
- Ma et al, Serodiagnosis of Lyme Borreliosis by Western Immunoblot: Reactivity of Various Significant Antibodies against Borrelia burgdorferi. Journal of Clinical Microbiology, Feb. 1992, p. 370-376.
- Variable manifestations, diverse seroreactivity and post-treatment persistence in non-human primates exposed to Borrelia burgdorferi by tick feeding. Monica E. Embers, et al. PLoS December 13, 2017. https://doi.org/10.1371/journal.pone.0189071
Regarding the symptoms described in case 1, he meets the prior criteria we discussed in Shadick, Steere and our own published work. We therefore do not agree with your assessment regarding his testing and symptoms. He also had a positive Babesia FISH post therapy with DDD CT, and since the patient never had a blood transfusion or organ transplant (the only other way to contract Babesia, is from maternal-fetal transmission), this also confirms his exposure to ticks and tick-borne infections.
- “Case 2: This patient had decades of symptoms, none of which were typical of Lyme or other tick-borne diagnoses, had no described exposure, and had elevated total serum IgM. Diagnosis was based on Lyme “ELISA + [1.20 IgG/IgM serology, normal < 1.0], CDC positive IgM Western Blot, serum Borrelia burgdorferi PCR +), Babesiosis (positive Babesia microti titer 1:160), Ehrlichiosis (Ehrlichia chaffeensis, human monocytic ehrlichia [HME] titers 1:160 +)”. In a patient with longstanding symptoms a low positive Lyme ELISA supported only by a positive IgM blot has an extremely high negative predictive value for the diagnosis of Lyme. Similarly, the other positive serologies are likely an artefact of the elevated total IgM. The significance of the positive serum PCR cannot be assessed without knowing which lab performed the test and that lab’s experience with false positives, as this finding is extraordinarily rare outside acute disease.
Response: Regarding case 2, we do not agree with your assessment for the following reasons: As we discussed in the scientific review of testing listed above, this patient has multiple markers of both exposure to borrelia and active Lyme (ELISA, CDC positive IgM blot and PCR +), as well as markers of multiple other tick-borne diseases. The significance of a positive IgM Western blot in late Lyme disease, was published in 2019 in our Precision Medicine paper: Horowitz, R.I.; Freeman, P.R. Precision Medicine: retrospective chart review and data analysis of 200 patients on dapsone combination therapy for chronic Lyme disease/post-treatment Lyme disease syndrome: part 1. International Journal of General Medicine 2019:12 101–119.
This was also highlighted in a Johns Hopkins University paper by Aucott and Rebman:
Rebman, A.W., Crowder, L.A., Kirkpatrick, A. et al. Characteristics of seroconversion and implications for diagnosis of post-treatment Lyme disease syndrome: acute and convalescent serology among a prospective cohort of early Lyme disease patients. Clin Rheumatol 34, 585–589 (2015). https://doi.org/10.1007/s10067-014-2706-z
“Two-tier serology is often used to confirm a diagnosis of Lyme disease. One hundred and four patients with physician diagnosed erythema migrans rashes had blood samples taken before and after 3 weeks of doxycycline treatment for early Lyme disease. Acute and convalescent serologies for Borrelia burgdorferi were interpreted according to the 2-tier antibody testing criteria proposed by the Centers for Disease Control and Prevention. Serostatus was compared across several clinical and demographic variables both pre- and post-treatment. Forty-one patients (39.4 %) were seronegative both before and after treatment. The majority of seropositive individuals on both acute and convalescent serology had a positive IgM western blot and a negative IgG western blot. IgG seroconversion on western blot was infrequent. Among the baseline variables included in the analysis, disseminated lesions (p < 0.0001), a longer duration of illness (p < 0.0001), and a higher number of reported symptoms (p = 0.004) were highly significantly associated with positive final serostatus, while male sex (p = 0.05) was borderline significant. This variability, and the lack of seroconversion in a subset of patients, highlights the limitations of using serology alone in identifying early Lyme disease. Furthermore, these findings underline the difficulty for rheumatologists in identifying a prior exposure to Lyme disease in caring for patients with medically unexplained symptoms or fibromyalgia-like syndromes.
Also, the symptoms described, are classic symptoms reported by Shadick, Steere and ourselves, as we discussed earlier. In our Precision Medicine paper, we found that 14.5% of our patients had a positive PCR for Borrelia burgdorferi despite long term treatment.
“Conclusion: Many of our patients infected with Lyme disease and associated coinfections had severe symptoms, often relapsed with commonly used therapies, and did not present with an EM rash nor meet the CDC two-tiered surveillance criteria. Almost two-thirds of patients had been exposed to between five and eight infections/coinfections and 14.5% of patients were PCR positive for B. burgdorferi despite seemingly “adequate” antibiotic therapy for months or years prior to DDS therapy (N=29, 14.5%). Evidence of persistent infection with HHV6, Bartonella, and/or Mycoplasma was also confirmed by PCR in several patients, although many in our study had evidence of other medical problems accounting for ongoing symptoms. These included associated immune dysfunction/immune deficiency, inflammation, environmental toxins with detoxification problems, GI problems, allergies, nutritional deficiencies, hormone, and autonomic nervous system dysregulation as well as sleep and psychiatric disorders in those suffering with post treatment Lyme symptoms.(151)
- “Case 3: This patient, who had potential exposure, is said to have had Lyme disease in the past, but it is not clear if this refers to a diagnosis prior to her current decade of symptoms; if it predates the 10 year symptomatology, no validating data or clinical description are provided. Her current symptoms began a decade before the current treatment, and IgG and IgM Western blots were said to be positive with no further detail. Since there is no description of a positive ELISA, one can assume it was negative, making the Western blot meaningless. We are told that on subsequent evaluation (line 421) “Her testing returned positive for Babesia and Borrelia burgdorferi” but no detail is provided, again making the statement uninterpretable. She is described as having extensive subsequent testing, but no results are provided; hence an assumption that it was all negative would be reasonable. None of her symptoms were diagnostic of Lyme disease and no data in support of other tick-borne illnesses is provided.
Response: As discussed above, patients can be negative on the ELISA test, and still show positive IgM and IgG Western blots. We provided extensive scientific evidence of this in our review of Lyme testing listed above. Similarly, the symptoms described in case 3, were classic symptoms seen in those diagnosed with long term Lyme disease and PTLDS, as per the published papers by Shadick, Steere and ourselves, referenced above.
Reviewer 2 Report
The authors have revealed three clinical cases where they have been subjected to the treatment of Double Dose Dapsone Combination and their results have been compared with other authors.
The article is of great interest to know the dosage used as well as the cycles for different patients to be treated
After to check the article , I find some improvements in the presentation of the article - As a proposal for improvement, it is proposed that the previously published study where placebos were used is presented, but not too much information is given to contrast the results shown. Since different clinical histories are presented in the studies shown -The authors should separate the cases under study from the introduction, it would be easier for the reader to differentiate the antecedents of the cases under study. -It would be good to move some paragraph of the discussion section to results, especially those where reference is made to "physical examination and blood tests" and the "Follow-up to the patients". -line 221-236, also I think that the result is not contrasted between patients, so it would be part of the description of the results.Author Response
Responses to Reviewer 2:
- “The authors have revealed three clinical cases where they have been subjected to the treatment of Double Dose Dapsone Combination and their results have been compared with other authors. The article is of great interest to know the dosage used as well as the cycles for different patients to be treated.
Response: Thank you for your comments. We added a figure in the paper, Figure 1. Double Dose Dapsone (DDD CT) Patient Care Plan which describes in detail the protocol, so that other health care providers can confirm and expand our results. See line 576 of the paper.
- “As a proposal for improvement, it is proposed that the previously published study where placebos were used is presented”
Response: We do not have placebo-controlled studies done on dapsone combination therapy. All prior dapsone studies that were published in the medical literature, are retrospective chart reviews among 300 patients with Lyme and associated tick-borne disorders.
Horowitz, R.I.; Freeman, P.R. Precision Medicine: retrospective chart review and data analysis of 200 patients on dapsone combination therapy for chronic Lyme disease/post-treatment Lyme disease syndrome: part 1. International Journal of General Medicine 2019:12 101–119. https://www.ncbi.nlm.nih.gov/pubmed/30863136
Horowitz RI, Freeman PR (2016) The Use of Dapsone as a Novel “Persister” Drug in the Treatment of Chronic Lyme Disease/Post Treatment Lyme Disease Syndrome. J Clin Exp Dermatol Res 7: 345. doi:10.4172/2155-9554.1000345
- “Since different clinical histories are presented in the studies shown -The authors should separate the cases under study from the introduction, it would be easier for the reader to differentiate the antecedents of the cases under study. -It would be good to move some paragraph of the discussion section to results, especially those where reference is made to "physical examination and blood tests" and the "Follow-up to the patients". -line 221-236, also I think that the result is not contrasted between patients, so it would be part of the description of the results”
Response: Thank you for your suggestions. Because we added 37 more patients, as per reviewer 3, results are now contrasted between patients, showing that active co-infections including Bartonella and Babesia, played a significant role in long term remission. Please see Table 1, line 610.
Since you asked us to move material from the introduction, we eliminated the following paragraph from the introduction on lines 91-line 97 of the old paper, and instead put in a Figure, which contains the relevant information.
Removed: “The dose of dapsone was doubled each week (week 1, 25 mg; week 2, 50 mg; week 3, 100 mg, weeks 4-8 100 mg BID), until reaching the final dose for double dose dapsone combination therapy (DDD CT). Nutritional support included N-acetyl cysteine 600 mg PO BID, alpha lipoic acid 600 mg PO BID, gradually increasing doses of glutathione up to 1000-2000 mg BID by the end of the first month, folinic acid (Leucovorin 25 mg PO BID month one, 25 mg TID month 2), L-methyl folate 15 mg PO BID, along with three biofilm agents (Stevia, oregano oil, Biocidin) and three probiotics (Theralac, Ultra Flora DF, saccharomyces boulardii).
Please see Figure 1, line 570.
Reviewer 3 Report
This manuscript describes the usage of double dose dapsone combination therapy to treat three patients suffering from the symptoms (often chronic and debilitating) associated with Lyme disease and coinfections (Bartonella, Babesia, Ehrlichia, Rickettsia, HHV-6, etc.). Tick-borne infections are often misdiagnosed/undiagnosed (for years), leading to undue suffering for patients and their families/caretakers. Moreover the failure of infectious disease physicians to acknowledge the necessity of varied treatments to combat Lyme disease and coinfections can be disappointing and harmful to suffering patients. Therefore, the data presented in this study is imperative for dissemination to practitioners. Lastly, the authors present a "whole organism/systems" approach by evaluating the health status of their patients beyond the simple presence/absence of Lyme disease and/or co-infections. This facet of tick-borne disease treatment is essential to properly combating the associated infection(s).
Major Comments:
-- Inclusion of data tables for each patient demonstrating changes in blood chemistry/profiles/values and for drug/herb/supplement administration during the course of treatment would make the data easier to visualize/analyze.
--Do the authors have additional cases of patients responding to DDDCT (formerly or currently)? This data might be worth adding to the Discussion.
Minor Comments:
--L48: Change "borrelia" to "Borrelia"
-- Throughout manuscript: italicize "in vitro" and "in vivo"
--L61: Change "Borrelia burgdorferi" to "B. burgdorgeri"
--L109: Change "Black" to "African American"
--L265: Change "white" to "Caucasian"
--L402: Change "white" to "Caucasian"
--L450: Change "chlamydia pneumonia" to "Chlamydia pneumoniae"
--L451: Change "Mycoplasma pneumonia" to "Mycoplasma pneumoniae"
--L452: Change "Mercury" to "mercury"
--L607: Is this sentence referring to B. burgdorferi? If so, it should be indicated within the sentence.
--615-616: Change "Borrelia burgdorferi" to "B. burgdorferi"
--L616: Change "borrelia" to "Borrelia"
--L619: Change "Hemophilus influenza" to "Haemophilus influenzae"
--L619: Change "Streptococcus pneumonia" to "Streptococcus pneumoniae"
--L620: Change "E. coli" to "Escherichia coli"
--L677: Change "Borrelia burgdorferi" to "B. burgdorferi"
Author Response
Responses to Reviewer 3:
- “Inclusion of data tables for each patient demonstrating changes in blood chemistry/profiles/values and for drug/herb/supplement administration during the course of treatment would make the data easier to visualize/analyze”.
Response: As per your suggestion, we included a figure in the paper, which clearly outlines the course of treatment using the different medications, herbs, and supplements. See Figure 1. Double Dose Dapsone (DDD CT) Patient Care Plan. Line 570.
The chemistry values are reported in each case (i.e. patient 1, lines 226-231, “Baseline white cell counts (WBC) of 3.9-4.3 x 10 E3/uL remained stable during treatment and hemoglobin/hematocrit levels (H/H) of 14.2/43.5 dropped to 12.2/37.9 during month two (a 2 gram drop in hemoglobin) with methemoglobin levels rising from 2.6% month one, to a maximum of 4.4% month two, which subsequently decreased to 3.9%, and 1.2% by the end of therapy (normal levels of methemoglobin range between 0 and 2.9%)”. Similar values are seen in each case study.
In the discussion section, lines 779-800, we discuss the changes in chemistry profiles “Dapsone, also has 4 common side effects, described as ‘Do No H.A.R.M.’, i.e., Herxheimer reactions (due to increased inflammatory cytokine production), Anemia (secondary to inhibition of folic acid metabolism, or hemolysis due to G-6-P-D deficiency), Rashes (due to sulfa sensitivity) and Methemoglobinemia (due to increased oxidative stress and diminished oxygen carrying capacity) [14] [118] [119] [120]. Although some of these symptoms were seen in our patients undergoing treatment with DDD CT (Herxheimer reactions, anemia, mild elevations in methemoglobin), adverse side effects were minimized by ruling out G-6-P-D deficiency, using high dose folic acid therapy with folinic acid (50-75 mg/day) and L-methyl folate (30-45 mg/day), as well as administering glutathione precursors (NAC 600 mg BID), alpha lipoic acid (ALA, 600 mg BID) and glutathione (GSH, 1000 mg BID) with methylene blue 50 mg BID as needed. Any decrease in red cell counts or significant anemia secondary to dapsone resolved in all patients within 1-2 months of stopping DDD CT while remaining on folic acid supplementation, and none of our patients developed rashes, nor significantly elevated levels of methemoglobin. Use of NAC, ALA and GSH helped to decrease oxidative stress, support detoxification and minimize the risk of methemoglobinemia [121] [122] [123], while doses of glutathione were increased to 2000 mg QD or BID along with alkalization (using sodium bicarbonate or fresh squeezed citrus) for Herxheimer reactions and/or any increased levels of methemoglobin [124] [76] [125]. Methylene blue can also be given orally to mitigate and rapidly reduce methemoglobinemia [120]. It was only necessary in one of our 3 patients reported here, although in other chronically ill Lyme-MSIDS patients given dapsone at 100 mg or higher [14], oral methylene blue was occasionally needed, and effective in keeping methemoglobin levels below 5%, allowing continuation of therapy”.
- “Do the authors have additional cases of patients responding to DDDCT (formerly or currently)? This data might be worth adding to the Discussion”
Response: As per your suggestion, we added an additional 37 patients to the paper, in a retrospective chart review. Please see Table 1, Co-infection Status and Treatment Response in 40 Patients on DDD CT. Line 610. This extra data helps to strengthen the paper and its conclusions, extends out the patient population to include those with PTLDS, and highlights the importance of active co-infections in those with Lyme disease. Thank you for asking us to do this, we feel the paper is much stronger with this extra information.
- Minor Comments:
- --L48: Change "borrelia" to "Borrelia". This was done on line 51 of the new manuscript.
- -- Throughout manuscript: italicize "in vitro" and "in vivo". This was done on lines 60, 724, 750, 769
- --L61: Change "Borrelia burgdorferi" to "B. burgdorgeri". Done, line 63-64
- --L109: Change "Black" to "African American". Done, line 107
- --L265: Change "white" to "Caucasian" Done, line 263
- --L402: Change "white" to "Caucasian" Done, line 400
- --L450: Change "chlamydia pneumonia" to "Chlamydia pneumoniae". Done, line 448
- --L451: Change "Mycoplasma pneumonia" to "Mycoplasma pneumoniae". Done, line 449
- --L452: Change "Mercury" to "mercury" Done, line 450
- --L607: Is this sentence referring to B. burgdorferi? If so, it should be indicated within the sentence. Yes, we changed it, on line 699; Good pick up.
- --615-616: Change "Borrelia burgdorferi" to "B. burgdorferi". Done, line 707
- --L616: Change "borrelia" to "Borrelia". Done, line 708
- --L619: Change "Hemophilus influenza" to "Haemophilus influenzae". Done, line 711
- --L619: Change "Streptococcus pneumonia" to "Streptococcus pneumoniae". Done, line 711
- --L620: Change "E. coli" to "Escherichia coli" Done, line 712
- --L677: Change "Borrelia burgdorferi" to "B. burgdorferi" Done, line 678
Reviewer 4 Report
In the presented manuscript authors described three patients with multi-year histories of relapsing and remitting Lyme-disease and associated co-infections. The cases are clearly presented with the valuable discussion. I suggest some minor revisions, mainly corrections of the pathogen’s names.
Specific comments
Lines 30-31: please correct "... range of bacteria(i.e. Borreliaspp., Rickettsia spp., Francisella tularensis), ...
"Line 34: please correct Borreliaspp.
Line 41: "Post-Treatment LymeDisease Syndrome"should be deleted, abbreviation is explained above (line 27).
Line 148: please correct human herpesvirus 6, Epstein-Barr virus, cytomegalovirus (virus names should not be written in italic).
Line 152: Please correct Rickettsia rickettsii
Line 153: please correct Coxiella burnetii
Line 239: please correct Bartonella quintana
Line 271: please correct human monocytic ehrlichiosis
Line 272: please correct human granulocytic anaplasmosis
Line 274: please correct Mycoplasma pnaumoniae
Line 450: please correctChlamydia pneumoniae
Line 451: please correct Mycoplasma pneumoniae
Line 589: please correct Mycoplasma spp.
Line 590: human herpesvirus 6 should be written in a lowercase letter, please correct.
Line 601:please correct Mycoplasma
Line 619: please correct Haemophilus influenzaeand Streptococcus pneumoniae.
References
Please check references 4, 10, 49, 51 (journal names).
Ref. 107: Authors are missing, please correct.
Ref. 135: Journal name is missing, please correct.
Author Response
Reviewer 4: In the presented manuscript authors described three patients with multi-year histories of relapsing and remitting Lyme-disease and associated co-infections. The cases are clearly presented with the valuable discussion. I suggest some minor revisions, mainly corrections of the pathogen’s names.
Specific comments
Lines 30-31: please correct "... range of bacteria (i.e. Borrelia spp., Rickettsia spp., Francisella tularensis), ...Done, lines 33-34
"Line 34: please correct Borreliaspp. Done, line 37
Line 41: "Post-Treatment LymeDisease Syndrome"should be deleted, abbreviation is explained above (line 27). Done, line 44
Line 148: please correct human herpesvirus 6, Epstein-Barr virus, cytomegalovirus (virus names should not be written in italic). Done, line 146
Line 152: Please correct Rickettsia rickettsii Done, line 150
Line 153: please correct Coxiella burnetii Done, line 151
Line 239: please correct Bartonella quintana Done, line 237
Line 271: please correct human monocytic ehrlichiosis Done, line 269
Line 272: please correct human granulocytic anaplasmosis Done, line 270
Line 274: please correct Mycoplasma pnaumoniae Done, line 272
Line 450: please correct Chlamydia pneumoniae Done, line 448
Line 451: please correct Mycoplasma pneumoniae Done, line 449
Line 589: please correct Mycoplasma spp. Done, line 680
Line 590: human herpesvirus 6 should be written in a lowercase letter, please correct. Done, line 681
Line 601: please correct Mycoplasma Done, line 692
Line 619: please correct Haemophilus influenzaeand Streptococcus pneumoniae. Done, line 711
References
Please check references 4, 10, 49, 51 (journal names).
Ref. 107: Authors are missing, please correct.
Ref. 135: Journal name is missing, please correct.
Done. We went through all of the references and corrected the ones highlighted. These were references 4, 10, 49, 51, 107, 108 (fixed, as our article was published), 135
Round 2
Reviewer 1 Report
As explained on the initial review, none of the 3 patients meet appropriate diagnostic criteria for Lyme disease. The authors have not provided any further information that disproves this conclusion